# Exactly Computing the Local Lipschitz Constant of ReLU networks

**Matt Jordan**
UT Austin
mjordan@cs.utexas.edu

**Alexandros G. Dimakis**
UT Austin
dimakis@austin.utexas.edu

## Abstract

The local Lipschitz constant of a neural network is a useful metric with applications in robustness, generalization, and fairness evaluation. We provide novel analytic results relating the local Lipschitz constant of nonsmooth vector-valued functions to a maximization over the norm of the generalized Jacobian. We present a sufficient condition for which backpropagation always returns an element of the generalized Jacobian, and reframe the problem over this broad class of functions. We show strong inapproximability results for estimating Lipschitz constants of ReLU networks, and then formulate an algorithm to compute these quantities exactly. We leverage this algorithm to evaluate the tightness of competing Lipschitz estimators and the effects of regularized training on the Lipschitz constant.

## 1 Introduction

We are interested in computing the Lipschitz constant of neural networks with ReLU activations. Formally, for a network $f$ with multiple inputs and outputs, we are interested in the quantity

$$\sup_{x \neq y} \frac{||f(x) - f(y)||_\beta}{||x - y||_\alpha}. \tag{1}$$

We allow the norm of the numerator and denominator to be arbitrary and further consider the case where $x, y$ are constrained in an open subset of $\mathbb{R}^n$ leading to the more general problem of computing the *local* Lipschitz constant.

Estimating or bounding the Lipschitz constant of a neural network is an important and well-studied problem. For the Wasserstein GAN formulation [1] the discriminator is required to have a bounded Lipschitz constant, and there are several techniques to enforce this [1–3]. For supervised learning Bartlett et al. [4] have shown that classifiers with lower Lipschitz constants have better generalization properties. It has also been observed that networks with smaller gradient norms are more robust to adversarial attacks. Bounding the (local) Lipschitz constant has been used widely for certifiable robustness against targeted adversarial attacks [5–7]. Lipschitz bounds under fair metrics may also be used as a means to certify the individual fairness of a model [8, 9].

The Lipschitz constant of a function is fundamentally related to the supremal norm of its Jacobian matrix. Previous work has demonstrated the relationship between these two quantities for functions that are scalar-valued and smooth [10, 11]. However, neural networks used for multi-class classification with ReLU activations do not meet either of these assumptions. We establish an analytical result that allows us to formulate the local Lipschitz constant of a vector-valued nonsmooth function as an optimization over the generalized Jacobian. We access the generalized Jacobian by means of the chain rule. As we discuss, the chain rule may produce incorrect results [12] for nonsmooth functions, even ReLU networks. To address this problem, we present a sufficient condition over the parameters of a ReLU network such that the chain rule always returns an element of the generalized Jacobian, allowing us to solve the proposed optimization problem.

Exactly computing Lipschitz constants of scalar-valued neural networks under the $\ell_2$ norm was shown to be NP-hard [13]. In this paper we establish strong inapproximability results showing that it is hard to even approximate Lipschitz constants of scalar-valued ReLU networks, for $\ell_1$ and $\ell_\infty$ norms.

A variety of algorithms exist that estimate Lipschitz constants for various norms. To the best of our knowledge, none of these techniques are exact: they are either upper bounds, or heuristic estimators with no provable guarantees. In this paper we present the first technique to provably *exactly* compute Lipschitz constants of ReLU networks under the $\ell_1, \ell_\infty$ norms. Our method is called LipMIP and relies on Mixed-Integer Program (MIP) solvers. As expected from our hardness results, our algorithm runs in exponential time in the worst case. At any intermediate time our algorithm may be stopped early to yield valid upper bounds.

We demonstrate our algorithm on various applications. We evaluate a variety of Lipschitz estimation techniques to definitively evaluate their relative error compared to the true Lipschitz constant. We apply our algorithm to yield reliable empirical insights about how changes in architecture and various regularization schemes affect the Lipschitz constants of ReLU networks.

Our contributions are as follows:

- We present novel analytic results connecting the Lipschitz constant of an arbitrary, possibly nonsmooth, function to the supremal norm of generalized Jacobians.
- We present a sufficient condition for which the chain rule will always yield an element of the generalized Jacobian of a ReLU network.
- We show that that it is provably hard to approximate the Lipschitz constant of a network to within a factor that scales almost linearly with input dimension.
- We present a Mixed-Integer Programming formulation (LipMIP) that is able to exactly compute the local Lipschitz constant of a scalar-valued ReLU network over a polyhedral domain.
- We analyze the efficiency and accuracy of LipMIP against other Lipschitz estimators. We provide experimental data demonstrating how Lipschitz constants change under training.

## 2    Gradient Norms and Lipschitz Constants

First we define the problem of interest. There have been several recent papers that leverage an analytical result relating the Lipschitz constant of a function to the maximal dual norm of its gradient [6, 10, 14]. This analytical result is limited in two aspects: namely it only applies to functions that are both scalar-valued and continuously differentiable. Neural networks with ReLU nonlinearities are nonsmooth and for multi-class classification or unsupervised learning settings, typically not scalar-valued. To remedy these issues, we will present a theorem relating the Lipschitz constant to the supremal norm of an element of the generalized Jacobian. We stress that this analytical result holds for all Lipschitz continuous functions, though we will only be applying this result to ReLU networks in the sequel.

The quantity we are interested in computing is defined as follows:

**Definition 1** *The **local** $(\alpha, \beta)$-**Lipschitz constant** of a function $f : \mathbb{R}^d \to \mathbb{R}^m$ over an open set $\mathcal{X} \subseteq \mathbb{R}^d$ is defined as the following quantity:*

$$L^{(\alpha,\beta)}(f, \mathcal{X}) := \sup_{x,y \in \mathcal{X}} \frac{||f(y) - f(x)||_\beta}{||x - y||_\alpha} \qquad (x \neq y) \tag{2}$$

*And if $L^{(\alpha,\beta)}(f, \mathcal{X})$ exists and is finite, we say that $f$ is $(\alpha, \beta)$-locally Lipschitz over $\mathcal{X}$.*

If $f$ is scalar-valued, then we denote the above quantity $L^\alpha(f, \mathcal{X})$ where $|| \cdot ||_\beta = | \cdot |$ is implicit. For smooth, scalar-valued $f$, it is well-known that

$$L^\alpha(f, \mathcal{X}) = \sup_{x \in \mathcal{X}} ||\nabla f(x)||_{\alpha^*}, \tag{3}$$

where $||z||_{\alpha^*} := \sup_{||y||_\alpha \leq 1} y^T z$ is the dual norm of $|| \cdot ||_\alpha$ [10, 11]. We seek to extend this result to be applicable to vector-valued nonsmooth Lipschitz continuous functions. As the Jacobian is not

well-defined everywhere for this class of functions, we recall the definition of Clarke's generalized Jacobian [15]:

**Definition 2** *The (Clarke)* ***generalized Jacobian*** *of $f$ at $x$, denoted $\delta_f(x)$, is the convex hull of the set of limits of the form $\lim_{i \to \infty} \nabla f(x_i)$ for any sequence $(x_i)_{i=1}^{\infty}$ such that $\nabla f(x_i)$ is well-defined and $x_i \to x$.*

Informally, $\delta_f(x)$ may be viewed as the convex hull of the Jacobian of nearby differentiable points. We remark that for smooth functions, $\delta_f(x) = \{\nabla f(x)\}$ for all $x$, and for convex nonsmooth functions, $\delta_f(\cdot)$ is the subdifferential operator.

The following theorem relates the norms of the generalized Jacobian to the local Lipschitz constant.

**Theorem 1** *Let $||\cdot||_\alpha, ||\cdot||_\beta$ be arbitrary convex norms over $\mathbb{R}^d, \mathbb{R}^m$ respectively, and let $f : \mathbb{R}^d \to \mathbb{R}^m$ be $(\alpha, \beta)$-Lipschitz continuous over an open set $\mathcal{X}$. Then the following equality holds:*

$$L^{(\alpha,\beta)}(f, \mathcal{X}) = \sup_{G \in \delta_f(\mathcal{X})} ||G^T||_{\alpha,\beta} \tag{4}$$

*where $\delta_f(\mathcal{X}) := \{G \in \delta_f(x) \mid x \in \mathcal{X}\}$ and $||M||_{\alpha,\beta} := \sup_{||v||_\alpha \leq 1} ||Mv||_\beta$.*

This result relies on the fact that Lipschitz continuous functions are differentiable almost everywhere (Rademacher's Theorem). As desired our result recovers equation 3 for scalar-valued smooth functions. Developing techniques to optimize the right-hand-side of equation 4 will be the central algorithmic focus of this paper.

## 3 ReLU Networks and the Chain Rule

Theorem 1 relates the Lipschitz constant to an optimization over generalized Jacobians. Typically we access the Jacobian of a function through backpropagation, which is simply an efficient implementation of the familiar chain rule. However the chain rule is only provably correct for functions that are compositions of continuously differentiable functions, and hence does not apply to ReLU networks [12]. In this section we will provide a sufficient condition over the parameters of a ReLU network such that any standard implementation of the chain rule will always yield an element of the generalized Jacobian.

**The chain rule for nonsmooth functions:** To motivate the discussion, we turn our attention to neural networks with ReLU nonlinearities. We say that a function is a ReLU network if it may be written as a composition of affine operators and element-wise ReLU nonlinearities, which may be encoded by the following recursion:

$$f(x) = c^T \sigma(Z_d(x)) \qquad Z_i(x) = W_i \sigma(Z_{i-1}(x)) + b_i \qquad Z_0(x) := x \tag{5}$$

where $\sigma(\cdot)$ here is the ReLU operator applied element-wise. We present the following example where the chain rule yields a result not contained in the generalized Jacobian. The univariate identity function may be written as $I(x) := 2x - \sigma(x) + \sigma(-x)$. Certainly at every point $x$, $\delta_I(x) = \{1\}$. However as Pytorch's automatic differentiation package defines $\sigma'(0) = 0$, Pytorch will compute $I'(0)$ as 2 [16]. Indeed, this is exactly the case where naively replacing the feasible set $\delta_f(\mathcal{X})$ in Equation 4 by the set of Jacobians returned by the chain rule will yield an incorrect calculation of the Lipschitz constant. To correctly relate the set of generalized Jacobians to the set of elements returnable by an implementation of the chain rule, we introduce the following definition:

**Definition 3** *Consider any implementation of the chain rule which may arbitrarily assign any element of the generalized gradient $\delta_\sigma(0)$ for each required partial derivative $\sigma'(0)$. We define the set-valued function $\nabla^\# f(\cdot)$ as the collection of answers yielded by **any** such chain rule.*

The subdifferential of the ReLU function at zero is the closed interval $[0, 1]$, so the chain rule as implemented in PyTorch and Tensorflow will yield an element contained in $\nabla^\# f(\cdot)$. Our goal will be to demonstrate that, for a broad class of ReLU networks, the feasible set in Equation 4 may be replaced by the set $\{G \in \nabla^\# f(x) \mid x \in \mathcal{X}\}$.

**General Position ReLU Networks:** Taking inspiration from hyperplane arrangements, we refer to this sufficient condition as *general position*. Letting $f : \mathbb{R}^d \to \mathbb{R}^m$ be a ReLU network with $n$ neurons, we can define the function $g_i(x) : \mathbb{R}^d \to \mathbb{R}$ for all $i \in [n]$ as the input to the $i^{th}$ ReLU of $f$ at $x$. Then we consider the set of inputs for which each $g_i$ is identically zero: we refer to the set $K_i := \{x \mid g_i(x) = 0\}$ as the $i^{th}$ *ReLU kernel* of $f$. We say that a polytope $P$ is $k$-dimensional if the affine hull of $P$ has dimension exactly $k$. Then we define general position ReLU networks as follows:

**Definition 4** *We say that a ReLU network with $n$ neurons is in **general position** if, for every subset of neurons $S \subseteq [n]$, the intersection $\cap_{i \in S} K_i$ is a finite union of $(d - |S|)$-dimensional polytopes.*

We emphasize that this definition requires that particular ReLU kernel is a finite union of $(d-1)$-dimensional polytopes, i.e. the 'bent hyperplanes' referred to in [17]. For a general position neural net, no $(d+1)$ ReLU kernels may have a nonempty intersection. We now present our theorem on the correctness of chain rule for general position ReLU networks.

**Theorem 2** *Let $f$ be a general position ReLU network, then for every $x$ in the domain of $f$, the set of elements returned by the generalized chain rule is exactly the generalized Jacobian:*

$$\nabla^{\#} f(x) = \delta_f(x) \tag{6}$$

In particular this theorem implies that, for general position ReLU nets,

$$L^{(\alpha,\beta)}(f, \mathcal{X}) = \sup_{G \in \nabla^{\#} f(\mathcal{X})} ||G^T||_{\alpha,\beta} \tag{7}$$

We will develop algorithms to solve this optimization problem predicated upon the assumption that a ReLU network is in general position. As shown by the following theorem, almost every ReLU network satisfies this condition.

**Theorem 3** *The set of ReLU networks not in general position has Lebesgue measure zero over the parameter space.*

## 4 Inapproximability of the Local Lipschitz Constant

In general, we seek algorithms that yield estimates of the Lipschitz constant of ReLU networks with provable guarantees. In this section we will address the complexity of Lipschitz estimation of ReLU networks. We show that under mild complexity theoretic assumptions, no deterministic polynomial time algorithm can provably return a tight estimate of the Lipschitz constant of a ReLU network

Extant work discussing the complexity of Lipschitz estimation of ReLU networks has only shown that computing $L^2(f, \mathbb{R}^d)$ is NP-hard [13]. This does not address the question of whether efficient approximation algorithms exist. We relate this problem to the problem of approximating the maximum independent set of a graph. Maximum independent set is one of the hardest problems to approximate: if $G$ is a graph with $d$ vertices, then assuming the Exponential Time Hypothesis[1], it is hard to approximate the maximum independent set of $G$ with an approximation ratio of $\Omega(d^{1-c})$ for any constant $c$. Our result achieves the same inapproximability result, where $d$ here refers to the encoding size of the ReLU network, which scales at least linearly with the input dimension and number of neurons.

**Theorem 4** *Let $f$ be a scalar-valued ReLU network, not necessarily in general position, taking inputs in $\mathbb{R}^d$. Then assuming the exponential time hypothesis, there does not exist a polynomial-time approximation algorithm with ratio $\Omega(d^{1-c})$ for computing $L^{\infty}(f, \mathcal{X})$ and $L^1(f, \mathcal{X})$, for any constant $c > 0$.*

## 5 Computing Local Lipschitz Constants With Mixed-Integer Programs

The results of the previous section indicate that one cannot develop any polynomial-time algorithm to estimate the local Lipschitz constant of ReLU network with nontrivial provable guarantees. Driven

by this negative result, we can instead develop algorithms that exactly compute this quantity but do not run in polynomial time in the worst-case. Namely we will use a mixed-integer programming (MIP) framework to formulate the optimization problem posed in Equation 7 for general position ReLU networks. For ease of exposition, we will consider scalar-valued ReLU networks under the $\ell_1, \ell_\infty$ norms, thereby using MIP to exactly compute $L^1(f, \mathcal{X})$ and $L^\infty(f, \mathcal{X})$. Our formulation may be extended to vector-valued networks and a wider variety of norms, which we will discuss in the supplementary.

While mixed-integer programming requires exponential time in the worst-case, implementations of mixed-integer programming solvers typically have runtime that is significantly lower than the worst-case. Our algorithm is unlikely to scale to massive state-of-the-art image classifiers, but we nevertheless argue the value of such an algorithm in two ways. First, it is important to provide a ground-truth as a frame of reference for evaluating the relative error of alternative Lipschitz estimation techniques. Second, an algorithm that provides provable guarantees for Lipschitz estimation allows one to make accurate claims about the properties of neural networks. We empirically demonstrate each of these use-cases in the experiments section.

We state the following theorem about the correctness of our MIP formulation and will spend the remainder of the section describing the construction yielding the proof.

**Theorem 5** *Let $f : \mathbb{R}^d \to \mathbb{R}$ be a general position ReLU network and let $\mathcal{X}$ be an open set that is the neighborhood of a bounded polytope in $\mathbb{R}^d$. Then there exists an efficiently-encodable mixed-integer program whose optimal objective value is $L^\alpha(f, \mathcal{X})$, where $|| \cdot ||_\alpha$ is either the $\ell_1$ or $\ell_\infty$ norm.*

**Mixed-Integer Programming:** Mixed-integer programming may be viewed as the extension of linear programming where some variables are constrained to be integral. The feasible sets of mixed-integer programs, may be defined as follows:

**Definition 5** *A **mixed-integer polytope** is a set $M \subseteq \mathbb{R}^n \times \{0,1\}^m$ that satisfies a set of linear inequalities:*

$$M := \{(x, a) \subseteq \mathbb{R}^n \times \{0,1\}^m \mid Ax + Ba \le c\} \tag{8}$$

Mixed-integer programming then optimizes a linear function over a mixed-integer polytope.

From equation 7, our goal is to frame $\nabla^\# f(\mathcal{X})$ as a mixed-integer polytope. More accurately, we aim to frame $\{||G^T||_\alpha \mid G \in \nabla^\# f(\mathcal{X})\}$ as a mixed-integer polytope. The key idea for how we do this is encapsulated in the following example. Suppose $\mathcal{X}$ is some set and we wish to solve the optimization problem $\max_{x \in \mathcal{X}} (g \circ f)(x)$. Letting $\mathcal{Y} := \{f(x) \mid x \in \mathcal{X}\}$ and $\mathcal{Z} := \{g(y) \mid y \in \mathcal{Y}\}$, we see that

$$\max_{x \in \mathcal{X}} (g \circ f)(x) = \max_{y \in \mathcal{Y}} g(y) = \max_{z \in \mathcal{Z}} z \tag{9}$$

Thus, if $\mathcal{X}$ is a mixed-integer polytope, and $f$ is such that $f(\mathcal{X})$ is also a mixed-integer polytope and similar for $g$, then the optimization problem may be solved under the MIP framework.

From the example above, it suffices to show that $\nabla^\# f(\cdot)$ is a composition of functions $f_i$ with the property that $f_i$ maps mixed-integer polytopes to mixed-integer polytopes without blowing up in encoding-size. We formalize this notion with the following definition:

**Definition 6** *We say that a function $g$ is **MIP-encodable** if, for every mixed-integer polytope $M$, the image of $M$ mapped through $g$ is itself a mixed-integer polytope.*

As an example, we show that the affine function $g(x) := Dx + e$ is MIP-encodable, where $g$ is applied only to the continuous variables. Consider the canonical mixed-integer polytope $M$ defined in equation 8, then $g(M)$ is the mixed-integer polytope over the existing variables $(x, a)$, with the dimension lifted to include the new continuous variable $y$ and a new equality constraint:

$$g(M) := \{(y, a) \mid (Ax + Ba \le c) \land (y = Dx + e)\}. \tag{10}$$

To represent $\{||G^T||_\alpha \mid x \in \nabla^\# f(\mathcal{X})\}$ as a mixed-integer polytope, there are two steps. First we must demonstrate a set of primitive functions such that $||\nabla^\# f(x)||_\alpha$ may be represented as a composition of these primitives, and then we must show that each of these primitives are MIP-encodable. In this sense, the following construction allows us to 'unroll' backpropagation into a mixed-integer polytope.

**MIP-encodable components of ReLU networks:** We introduce the following three primitive operators and show that $||\nabla^\# f||_\alpha$ may be written as a composition of these primitive operators. These operators are the affine, conditional, and switch operators, defined below:

**Affine operators:** For some fixed matrix $W$ and vector $b$, $A : \mathbb{R}^n \to \mathbb{R}^m$ is an affine operator if it is of the form $A(x) := Wx + b$.

The **conditional operator** $C : \mathbb{R} \to \mathcal{P}(\{0,1\})$ is defined as

$$C(x) = \begin{cases} \{1\} & \text{if } x > 0 \\ \{0\} & \text{if } x < 0 \\ \{0,1\} & \text{if } x = 0. \end{cases} \tag{11}$$

The **switch operator** $S : \mathbb{R} \times \{0,1\} \to \mathbb{R}$ is defined as

$$S(x, a) = x \cdot a. \tag{12}$$

Then we have the two following lemmas which suffice to show that $\nabla^\# f(\cdot)$ is a MIP-encodable function:

**Lemma 1** *Let $f$ be a scalar-valued general position ReLU network. Then $f(x)$, $\nabla^\# f(x)$, $|| \cdot ||_1$, and $|| \cdot ||_\infty$ may all be written as a composition of affine, conditional and switch operators.*

This is easy to see for $f(x)$ by the recurrence in Equation 5; indeed this construction is used in the MIP-formulation for evaluating robustness of neural networks [19–24]. For $\nabla^\# f$, one can define the recurrence:

$$\nabla^\# f(x) = W_1^T Y_1(x) \qquad Y_i(x) = W_{i+1}^T \text{Diag}(\Lambda_i(x))Y_{i+1}(x) \qquad Y_{d+1}(x) = c \tag{13}$$

where $\Lambda_i(x)$ is the conditional operator applied to the input to the $i^{th}$ layer of $f$. Since $\Lambda_i(x)$ takes values in $\{0,1\}^*$, $\text{Diag}(\Lambda(x))Y_{i+1}(x)$ is equivalent to $S(Y_{i+1}(x), \Lambda_i(x))$.

**Lemma 2** *Let $g$ be a composition of affine, conditional and switch operators, where global lower and upper bounds are known for each input to each element of the composition. Then $g$ is a MIP-encodable function.*

As we have seen, affine operators are trivially MIP-encodable. For the conditional and switch operators, global lower and upper bounds are necessary for MIP-encodability. Provided that our original set $\mathcal{X}$ is bounded, there exist several efficient schemes for propagating upper and lower bounds globally. Conditional and switch operators may be incorporated into the composition by adding only a constant number of new linear inequalities for each new variable. These constructions are described in full detail in the supplementary.

**Formulating LipMIP:** To put all the above components together, we summarize our algorithm. Provided a bounded polytope $\mathcal{P}$, we first compute global lower and upper bounds to each conditional and switch operator in the composition that defines $||\nabla^\# f(\cdot)||_\alpha$ by propagating the bounds of $\mathcal{P}$. We then iteratively move components of the composition into the feasible set as in Equation 9 by lifting the dimension of the feasible set and incorporating new constraints and variables. This yields a valid mixed-integer program which can be optimized by off-the-shelf solvers to yield $L^\alpha(f, \mathcal{X})$ for either the $\ell_1$ or $\ell_\infty$ norms.

**Extensions:** While our results focus on evaluating the $\ell_1$ and $\ell_\infty$ Lipschitz constants of scalar-valued ReLU networks, we note that the above formulation is easily extensible to vector-valued networks over a variety of norms. We present this formulation, including an application to untargeted robustness verification through the use of a novel norm in the supplementary. We also note that any convex relaxation of our formulation will yield a provable upper bound to the local Lipschitz constant. Mixed-integer programming formulations have natural linear programming relaxations, by relaxing each integral constraint to a continuous constraint. We denote this linear programming relaxation as LipLP. Most off-the-shelf MIP solvers may also be stopped early, yielding valid upper bounds for the Lipschitz constant.

# 6 Related Work

**Related Theoretical Work:** The analytical results in section 2 are based on elementary analytical techniques, where the formulation of generalized Jacobians is famously attributed to Clarke [15].

The problems with automatic differentiation over nonsmooth functions have been noted several times before [12, 25, 26]. In particular, in [12], the authors provide a randomized algorithm to yield an element of the generalized Jacobian almost surely. We instead present a result where the standard chain rule will return the correct answer everywhere for almost every ReLU network. The hardness of Lipschitz estimation was first proven by [13], and the only related inapproximability result is the hardness of approximating robustness to $\ell_1$-bounded adversaries in [6].

**Connections to Robustness Certification:**   We note the deep connection between certifying the robustness of neural networks and estimating the Lipschitz constant. Mixed-integer programming has been used to exactly certify the robustness of ReLU networks to adversarial attacks [19–24]. Broadly speaking, the mixed-integer program formulated in each of these works is the same formulation we develop to emulate the forward-pass of a ReLU network. Our work may be viewed as an extension of these techniques where we emulate the forward and backward pass of a ReLU network with mixed-integer programming, instead of just the forward pass. We also note that the subroutine we use for bound propagation is exactly the formulation of FastLip [6], which can be viewed as a form of reachability analysis, for which there is a deep body of work in the adversarial robustness setting [27–30].

**Lipschitz Estimation Techniques:**   There are many recent works providing techniques to estimate the local Lipschitz constant of ReLU networks. These can be broadly categorized by the guarantees they provide and the class of neural networks and norms they apply to. Extant techniques may either provide lower bounds, heuristic estimates [13, 14], or provable upper bounds [6, 10, 31] to the Lipschitz constant. These techniques may estimate $L^\alpha(f, \mathcal{X})$ for $|| \cdot ||_\alpha$ being an arbitrary $\ell_p$ norm [6, 13, 14], or only the $\ell_2$ norm [31]. Several of these techniques provide only global Lipschitz estimates [13, 31], where others are applicable to both local and global estimates [6, 10, 14]. Finally, some techniques are applicable to neural networks with arbitrary nonlinearities [13, 14], neural networks with only continuously differentiable nonlinearities [10], or just ReLU nonlinearities [6, 13]. We compare the performance of our proposed algorithm against the performance of several of these techniques in the experimental section.

# 7   Experiments

We have described an algorithm to exactly compute the Lipschitz constant of a ReLU network. We now demonstrate several applications where this technique has value. First we will compare the performance and accuracy of the techniques introduced in this paper to other Lipschitz estimation techniques. Then we will apply LipMIP to a variety of networks with different architectures and different training schemes to examine how these changes affect the Lipschitz constant. Full descriptions of the computing environment and experimental details are contained in the supplementary. We have also included extra experiments analyzing random networks, how estimation changes during training, and an application to vector-valued networks in the supplementary.

**Accuracy vs. Efficiency:** As is typical in approximation techniques, there is frequently a tradeoff between efficiency and accuracy. This is the case for Lipschitz estimation of neural nets. While ours is the first algorithm to provide quality guarantees about the returned estimate, it is worthwhile to examine how accurate the extant techniques for Lipschitz estimation are. We compare against the following estimation techniques: CLEVER [14], FastLip [6], LipSDP [31], SeqLip [13] and our MIP formulation (LipMIP) and its LP-relaxation (LipLP). We also provide the accuracy of a random lower-bounding technique where we report the maximum gradient dual norm over a random selection of test points (RandomLB) and a naive upper-bounding strategy (NaiveUB) where we report the product of the operator norm of each affine layer and scale by $\sqrt{d}$ due to equivalence of norms. In Table 1, we demonstrate the runtime and relative error of each considered technique. We evaluate each technique over the unit hypercube across random networks, networks trained on synthetic datasets, and networks trained to distinguish between MNIST 1's and 7's.

**Effect of Training On Lipschitz Constant:** As other techniques do not provide reliable estimates of the Lipschitz constant, we argue that these are insufficient for making broad statements about how the parameters or training scheme of a neural network affect the Lipschitz constant. In Figure 1 (left), we compare the returned estimate from a variety of techniques as a network undergoes training on a synthetic dataset. Notice how the estimates decrease in quality as training proceeds. On the

| | | Binary MNIST | | Synthetic Dataset | |
|---|---|---|---|---|---|
| **Method** | **Guarantee** | **Time (s)** | **Rel. Err.** | **Time (s)** | **Rel. Err.** |
| **RandomLB** | Lower | $0.334 \pm 0.019$ | $-41.96\%$ | $0.297 \pm 0.004$ | $-32.68\%$ |
| **CLEVER** | Heuristic | $20.574 \pm 4.320$ | $-36.97\%$ | $1.849 \pm 0.054$ | $+28.45\%$ |
| **LipMIP** | Exact | $69.187 \pm 70.114$ | $\mathbf{0.00}\%$ | $38.844 \pm 34.906$ | $\mathbf{0.00}\%$ |
| **LipLP** | Upper | $0.226 \pm 0.023$ | $+39.39\%$ | $0.030 \pm 0.002$ | $+362.43\%$ |
| **FastLip** | Upper | $0.002 \pm 0.000$ | $+63.41\%$ | $0.001 \pm 0.000$ | $+388.14\%$ |
| **LipSDP** | Upper | $20.570 \pm 2.753$ | $+113.92\%$ | $2.704 \pm 0.019$ | $+39.07\%$ |
| **SeqLip** | Heuristic | $0.022 \pm 0.005$ | $+119.53\%$ | $0.016 \pm 0.002$ | $+98.98\%$ |
| **NaiveUB** | Upper | $0.000 \pm 0.000$ | $+212.68\%$ | $0.000 \pm 0.000$ | $+996.96\%$ |

Table 1: Lipschitz Estimation techniques applied to networks of size [784, 20, 20, 20, 2] trained to distinguish MNIST 1's from 7's evaluated over $\ell_\infty$ balls of size 0.1, and networks of size [10, 20, 30, 20, 2] trained on synthetic datasets evaluated over the unit hypercube. Our method is the slowest, but provides a provably exact answer. This allows us to reliably gauge the accuracy and efficiency of the other techniques.

other hand, in Figure 1 (right), we use LipMIP to provide reliable insights as to how the Lipschitz constant changes as a neural network is trained on a synthetic dataset under various training schemes. A similar experiment where we vary network architecture is presented in the supplementary.

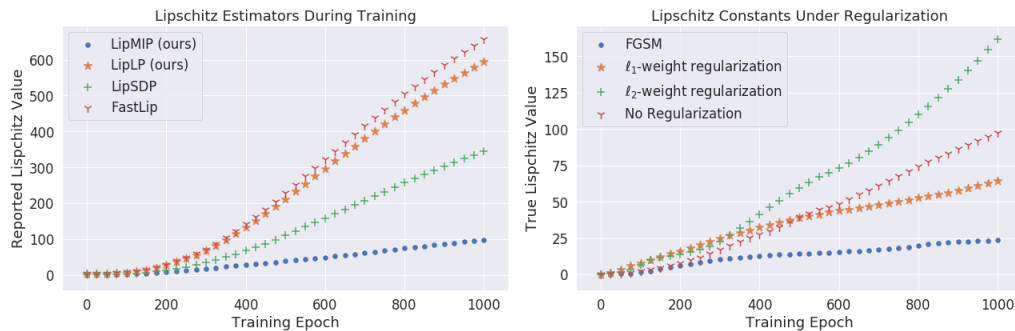

Figure 1: (Left): We plot how the bounds provided by Lipschitz estimators as we train a neural network on a synthetic dataset. We notice that as training proceeds, the absolute error of estimation techniques increases relative to the true Lipschitz constant computed with our method (blue dots). (Right): We plot the effect of regularization on Lipschitz constants during training. We fix a dataset and network architecture and train with different regularization methods and evaluate the Lipschitz constant with LipMIP. Observe that these values increase as training proceeds. Surprisingly, $\ell_2$ weight regularization increases the Lipschitz constant even compared to the no regularization baseline. Adversarial training (FGSM) is the most effective Lipschitz regularizer we found in this experiment.

## 8    Conclusion and Future Work

We framed the problem of local Lipschitz computation of a ReLU network as an optimization over generalized Jacobians, yielding an analytical result that holds for all Lipschitz continuous vector-valued functions. We further related this to an optimization over the elements returnable by the chain rule and demonstrated that even approximately solving this optimization problem is hard. We propose a technique to exactly compute this value using mixed integer programming solvers. Our exact method takes exponential time in the worst case but admits natural LP relaxations that trade-off accuracy for efficiency. We use our algorithm to evaluate other Lipschitz estimation techniques and evaluate how the Lipschitz constant changes as a network undergoes training or changes in architecture.

There are many interesting future directions. We have only started to explore relaxation approaches based on LipMIP and a polynomial time method that scales to large networks may be possible. The reliability of an exact Lipschitz evaluation technique may also prove useful in developing both new empirical insights and mathematical conjectures.

## Acknowledgments and Disclosure of Funding

This research has been supported by NSF Grants CCF 1763702,1934932, AF 1901292, 2008710, 2019844 research gifts by Western Digital, WNCG IAP, computing resources from TACC and the Archie Straiton Fellowship.

## 9 Broader Impact

As deep learning begins to see use in situations where safety or fairness are critical, it is increasingly important to have tools to audit and understand these models. The Lipschitz computation technique we have outlined in this work is one of these tools. As we have discussed, an upper bound on the Lipschitz constant of a model may be used to efficiently generate certificates of robustness against adversarial attacks. Lipschitz estimates have the advantage over other robustness certificates in that they may be used to make robustness claims about large subsets of the input space, rather than certifying that a particular input is robust against adversarial attacks. Lipschitz estimation, if com.putable with respect to fair metrics, may be utilized to generate certificates of individual fairness (see [8, 9] for examples of this formulation of fair metrics and individual fairness). Our approach is the first to provide a scheme for Lipschitz estimation with respect to arbitrary norms, which may include these fair metrics.

Exact verification of neural networks has the added benefit that we are guaranteed to be generate the correct answer and not just a sound approximation. We argue that a fundamental understanding of the behavior of these models needs to be derived from both theoretical results and accurate empirical validation. As we have demonstrated, our technique is able to provide accurate measurements of the Lipschitz constant of small-scale neural networks. The computational complexity of the problem suggests that such accurate measurements are not tractably attainable for networks with millions of hyperparameters. Our experiments demonstrate that our technique is scalable to networks large enough that insights may be drawn, such as claims about how regularized training affects the Lipschitz constant. Further, exact verification techniques may be used as benchmarks to verify the accuracy of the more efficient verification techniques. Future Lipschitz estimation techniques, assuming that they do not provide provable guarantees, will need to assert the accuracy of their reported answers: it is our hope that this will be empirically done by comparisons against exact verification techniques, where the accuracy claims may then be extrapolated to larger networks.

## Footnotes

[1]This states that 3SAT cannot be solved in sub-exponential time [18]. If true, this would imply $P \neq NP$.

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
