[Supplementary Material 1]

# Appendices

Contained in the supplementary are several standalone components. We have tried to include all relevant preliminary information where applicable. The proof of Theorem 1 is contained in Appendix A. The proofs of the statements in section 3 of the main paper are contained in Appendix B. The inapproximability results are presented in Appendix C. The formal construction and bound-propagation schemes of LipMIP are contained in Appendix D. Extension of LipMIP to general position vector-valued networks over a wider class of norms is contained in Appendix E. Appendix F has full experimental details and additional experiments.

## Contents

## A. Analytical proofs

We first start with formal definitions and known facts. We present our results in general for vector-valued functions, but we will make remarks about the implications for scalar-valued networks along the way.

### A.1. Definitions and Preliminaries

#### A.1.1. NORMS

As we will be frequently referring to arbitrary norms, we recall the formal definition:

**Definition A.1.** *A **norm** $||\cdot||$ over vector space $V$ is a nonnegative valued function that meets the following three properties:*

- ***Triangle Inequality:*** *For all $x, y \in V$, $||x + y|| \leq ||x|| + ||y||$*

- ***Absolute Homogeneity:*** *For all $x \in V$, and any field element $a$, $||ax|| = |a| \cdot ||x||$.*

- ***Point Separation:*** *If $||x|| = 0$, then $x = 0$, the zero vector of $V$.*

The most common norms are the $\ell_p$ norms over $\mathbb{R}^d$, with $||x||_p := \left(\sum_i |x_i|^p\right)^{1/p}$, though these are certainly not all possible norms over $\mathbb{R}^d$. We can also describe norms over matrices. One such norm that we frequently discuss is a norm over matrices in $\mathbb{R}^{m \times n}$ and is induced by norms over $\mathbb{R}^d$ and $\mathbb{R}^m$:

**Definition A.2.** *Given norm $||\cdot||_\alpha$ over $\mathbb{R}^d$, and norm $||\cdot||_\beta$ over $\mathbb{R}^m$, the matrix norm $||\cdot||_{\alpha,\beta}$ over $\mathbb{R}^{m \times n}$ is defined as*

$$||A||_{\alpha,\beta} := \sup_{||x||_\alpha \leq 1} ||Ax||_\beta = \sup_{x \neq 0} \frac{||Ax||_\beta}{||x||_\alpha} \tag{1}$$

A convenient way to keep the notation straight is that $A$, above, can be viewed as a linear operator which maps elements from a space which has norm $||\cdot||_\alpha$ to a space which has norm $||\cdot||_\beta$, and hence is equipped with the norm $||A||_{\alpha,\beta}$. As long as $||\cdot||_\alpha, ||\cdot||_\beta$ are norms, then $||\cdot||_{\alpha,\beta}$ is a norm as well in that the three properties listed above are satisfied.

Every norm induces a dual norm, defined as

$$||x||_* := \sup_{||y|| \leq 1} |\langle x, y \rangle| \tag{2}$$

Where the $\langle \cdot, \cdot \rangle$ is the standard inner product for vectors over $\mathbb{R}^d$ or matrices $\mathbb{R}^{m \times d}$. We note that if matrix $A$ is a row-vector, then $||A||_{\alpha,|\cdot|} = ||A||_{\alpha*}$ by definition.

We also have versions of Holder's inequality for arbitrary norms over $\mathbb{R}^d$:

**Proposition A.1.** *Let $||\cdot||_\alpha$ be a norm over $\mathbb{R}^d$, with dual norm $||\cdot||_{\alpha*}$. Then, for all $x, y \in \mathbb{R}^d$*

$$x^T y \leq ||x||_\alpha \cdot ||y||_{\alpha*} \tag{3}$$

*Proof.* Indeed, assuming WLOG that neither $x$ nor $y$ are zero, and letting $u = \frac{x}{||x||_\alpha}$, we have

$$x^T y = ||x||_\alpha \cdot u^T y \leq ||x||_\alpha \cdot \sup_{||u||_\alpha \leq 1} u^T y = ||x||_\alpha \cdot ||y||_{\alpha*} \tag{4}$$

$\square$

We can make a similar claim about the matrix norms defined above, $||\cdot||_{\alpha,\beta}$:

**Proposition A.2.** *Letting $||\cdot||_{\alpha,\beta}$ be a matrix norm induced by norms $||\cdot||_\alpha$ over $\mathbb{R}^d$, and $||\cdot||_\beta$ over $\mathbb{R}^m$, for any $A \in \mathbb{R}^{m \times n}$, $x \in \mathbb{R}^d$:*

$$||Ax||_\beta \leq ||A||_{\alpha,\beta} ||x||_\alpha \tag{5}$$

*Proof.* Indeed, assuming WLOG that $x$ is nonzero, letting $y = x/||x||_\alpha$ such that $||y||_\alpha = 1$, we have

$$||Ax||_\beta = ||x||_\alpha ||Ay||_\beta \leq \sup_{||y||_\alpha \leq 1} ||x||_\alpha ||Ay||_\beta = ||x||_\alpha ||A||_{\alpha,\beta} \tag{6}$$

$\square$

110 A.1.2. LIPSCHITZ CONTINUITY AND DIFFERENTIABILITY

111
112 When $f : \mathbb{R}^d \to \mathbb{R}^m$ is a vector-valued over some open set $\mathcal{X} \subseteq \mathbb{R}^d$ we say that it is $(\alpha, \beta)$-Lipschitz continuous if there
113 exists a constant $L$ for norms $||\cdot||_\alpha, ||\cdot||_\beta$ such that all $x, y \in \mathcal{X}$,

$$||f(x) - f(y)||_\beta \leq L \cdot ||x - y||_\alpha \tag{7}$$

116 Then the Lipschitz constant, $L^{(\alpha,\beta)}(f, \mathcal{X})$, is the infimum over all such $L$. Equivalently, one can define $L^{\alpha,\beta}(f, \mathcal{X})$ as

$$L^{(\alpha,\beta)}(f, \mathcal{X}) = \sup_{x,y \in \mathcal{X},; x \neq y} \frac{||f(x) - f(y)||_\beta}{||x - y||_\alpha} \tag{8}$$

121 We say that $f$ is differentiable at $x$ if there exists some linear operator $\nabla f(x) \in \mathbb{R}^{n \times m}$ such that

$$\lim_{h \to 0} \frac{f(x + h) - f(x) - \nabla f(x)^T h}{||h||} = 0 \tag{9}$$

126 A linear operator such that the above equation holds is defined as the Jacobian [1]

127 The directional derivative of $f$ along direction $v \in \mathbb{R}^d$ is defined as

$$d_v f(x) := \lim_{t \to 0} \frac{f(x + tv) - f(x)}{t} \tag{10}$$

131 Where we note that we are taking limits of a vector-valued function. We now add the following known facts:

- If $f$ is lipschitz continuous, then it is absolutely continuous.

- If $f$ is differentiable at $x$, all directional derivatives exist at $x$. The converse is not true, however.

- If $f$ is differentiable at $x$, then for any vector $v$, $d_v f(x) = \nabla f(x)^T v$.

- **(Rademacher's Theorem):** If $f$ is Lipschitz continuous, then $f$ is differentiable everywhere except for a set of measure zero, under the standard Lebesgue measure in $\mathbb{R}^d$ (Heinonen, 2005).

141 Finally we introduce some notational shorthand. Letting $f : \mathbb{R}^d \to \mathbb{R}^m$, be Lipschitz continuous and defined over an open
142 set $\mathcal{X}$, we denote Diff$(\mathcal{X})$ refer to the differentiable subset of $\mathcal{X}$. Let $\mathcal{D}$ be the set of $(x, v) \in \mathbb{R}^{2n}$ for which $d_v f(x)$ exists
143 and $x \in \mathcal{X}$. Additionally, let $\mathcal{D}_v$ be the set $\mathcal{D}_v = \{x \mid (x, v) \in \mathcal{D}\}$.

## A.2. Proof of Theorem 1

147 Now we can state our first lemma, which claims that for any norm, the maximal directional derivative is attained at a
148 differentiable point of $f$:

**Lemma A.1.** *For any $(\alpha, \beta)$ Lipschitz continuous function $f$, norm $||\cdot||_\beta$ over $\mathbb{R}^m$, any $v \in \mathbb{R}^d$, letting $\mathcal{D}_v := \{x \mid (x, v) \in \mathcal{D}\}$, we have:*

$$\sup_{x \in \mathcal{D}_v} ||d_v f(x)||_\beta \leq \sup_{x \in \text{Diff}(\mathcal{X})} ||\nabla f(x)^T v||_\beta \tag{11}$$

**Remark:** For scalar-valued functions and norm $||\cdot||_\alpha$ over $\mathbb{R}^d$, one can equivalently state that for all vectors $v$ with $||v||_\alpha = 1$:

$$\sup_{x \in \mathcal{D}_v} |d_v f(x)| \leq \sup_{x \in \text{Diff}(\mathcal{X})} ||\nabla f(x)||_{\alpha^*} \tag{12}$$

159 *Proof.* Essentially the plan is to say each of the following quantities are within $\epsilon$ of each other: $||d_v f(x)||_\beta$, the limit
160 definition of $||d_v f(x)||_\beta$, the limit definition of $||d_v f(x')||_\beta$ for nearby differentiable $x'$, and the norm of the gradient at $x'$
161 applied to the direction $v$.

We fix an arbitrary $v \in \mathbb{R}^d$. It suffices to show that for every $\epsilon > 0$, there exists some differentiable $x' \in \text{Diff}(\mathcal{X})$ such that $||\nabla f(x')^T \cdot v|| \geq \sup_{x \in \mathcal{D}_v} ||d_v f(x)|| - \epsilon$.

By the definition of sup, for every $\epsilon > 0$, there exists an $x \in \mathcal{D}_v$ such that

$$||d_v f(x)||_\beta \geq \sup_{y \in \mathcal{D}_v} ||\delta_f(y)||_\beta - \epsilon/4 \tag{13}$$

Then for all $\epsilon > 0$, by the limit definition of $d_v f(x)$ there exists a $\delta > 0$ such that for all $t$ with $|t| < \delta$

$$\left|\left| d_v f(x) - \left( \frac{f(x+tv) - f(x)}{||tv||_\alpha} \right) \right|\right|_\beta \leq \epsilon/4 \tag{14}$$

Next we note that, since lipschitz continuity implies absolute continuity of $f$, and $t$ is now a fixed constant, the function $h(x) := \frac{f(x)}{||tv||_\alpha}$ is absolutely continuous. Hence there exists some $\delta'$ such that for all $y \in \mathcal{X}$, $z$ with $||z||_\alpha \leq \delta'$

$$\frac{||f(y+z) - f(y)||_\beta}{||tv||_\alpha} = ||h(y+z) - h(y)||_\beta \leq \epsilon/4 \tag{15}$$

Hence, by Rademacher's theorem, there exists some differentiable $x'$ within a $\delta'$-neighborhood of $x$, such that both $\frac{||f(x') - f(x)||_\beta}{||tv||_\alpha} < \epsilon/4$ and $\frac{||f(x'+tv) - f(x+tv)||_\beta}{||tv||_\alpha} < \epsilon/4$, hence by the triangle inequality for $|| \cdot ||_\beta$

$$\left|\left| h(x+tv) - h(x) \right|\right|_\beta \leq \left|\left| h(x+tv) - h(x'+tv) \right|\right|_\beta + \left|\left| h(x) - h(x') \right|\right|_\beta + \left|\left| h(x'+tv) - h(x') \right|\right|_\beta \tag{16}$$

$$\leq \epsilon/2 + \left|\left| h(x'+tv) - h(x') \right|\right|_\beta$$

$$= \epsilon/2 + \frac{||f(x'+tv) - f(x')||_\beta}{||tv||_\alpha}$$

Combining equations 14 and 16 we have that

$$||d_v f(x)||_\beta \leq 3\epsilon/4 + \frac{||f(x'+tv) - f(x')||_\beta}{||tv||_\alpha} \tag{17}$$

Taking limits over $\delta \to 0$, we get that the final term in equation 17 becomes $3\epsilon/4 + ||d_v f(x')||_\beta$, which is equivalent to $3\epsilon/4 + ||\nabla f(x)^T v||_\beta$. Hence we have that

$$||\nabla f(x')^T \cdot v||_\beta \geq \sup_{x \in \mathcal{D}_v} ||d_v f(x)||_\beta - \epsilon \tag{18}$$

as desired, as our choice of $v$ was arbitrary.

$\square$

Now we can restate and prove our main theorem.

**Theorem A.1.** *Let $|| \cdot ||_\alpha$, $|| \cdot ||_\beta$ be arbitrary norms over $\mathbb{R}^d, \mathbb{R}^m$, and let $f : \mathbb{R}^d \to \mathbb{R}^m$ be locally $(\alpha, \beta)$-Lipschitz continuous over an open set $\mathcal{X}$. The following equality holds:*

$$L^{(\alpha,\beta)}(f, \mathcal{X}) = \sup_{G \in \delta_f(\mathcal{X})} ||G^T||_{\alpha,\beta} \tag{19}$$

**Remarks:** Before we proceed with the proof, we make some remarks. First, note that if $f$ is scalar-valued and continuously differentiable, then $\nabla f(x)^T$ is a row-vector, and $||\nabla f(x)^T||_{\alpha,\beta} = ||\nabla f(x)||_{\alpha^*}$, recovering the familiar known result. Second, to gain some intuition for this statement, consider the case where $f(x) = Ax + b$ is an affine function. Then $\nabla f(x)^T = A$, and by applying the theorem and leveraging the definition of $L^{(\alpha,\beta)}(f, \mathcal{X})$, we have

$$L^{(\alpha,\beta)}(f, \mathcal{X}) := \sup_{x \neq y \in \mathcal{X}} \frac{||A(x-y)||_\beta}{||x-y||_\alpha} = ||A||_{\alpha,\beta}, \tag{20}$$

where the last equality holds because $\mathcal{X}$ is open.

*Proof.* It suffices to prove the following equality:

$$L^{(\alpha,\beta)}(f,\mathcal{X}) = \sup_{x \in \text{Diff}(\mathcal{X})} ||\nabla f(x)^T||_{\alpha,\beta} \tag{21}$$

This follows naturally as if $x \in \text{Diff}(\mathcal{X})$ then $\delta_f(x) = \{\nabla f(x)\}$. On the other hand, if $x \notin \text{Diff}(\mathcal{X})$, then for every extreme point $G$ in $\delta_f(x)$, there exists an $x' \in \text{Diff}(\mathcal{X})$ such that $\nabla f(x') = G$ (by definition). As we seek to optimize over a norm, which is by definition convex, there exists an extreme point of $\delta_f(x)$ which attains the optimal value. Hence, we proceed by showing that Equation 21 holds.

We show that for all $x, y \in \mathcal{X}$ that $\frac{||f(x)-f(y)||_\beta}{||x-y||_\alpha}$ is bounded above by $\sup_{x \in \text{Diff}(\mathcal{X})} ||\nabla f(x)||_{\alpha,\beta}$. Then we will show the opposite inequality.

Fix any $x, y \in \mathcal{X}$, and note that since the dual of a dual norm is the original norm,

$$||f(x) - f(y)||_\beta = \sup_{||c||_{\beta^*} \leq 1} |c^T(f(x) - f(y))| \tag{22}$$

Moving the sup to the outside, we have

$$||f(x) - f(y)||_\beta = \sup ||c||_{\beta^*} |h_c(0) - h_c(1)| \tag{23}$$

for $h_c : \mathbb{R} \to \mathbb{R}$ defined as $h_c(t) := c^T f(x + t(y - x))$. Then certainly $h_c$ is lipschitz continous on the interval $[0, 1]$, and the limit $h'_c(t)$ exists almost everywhere, defined as

$$h'_c(t) := \lim_{\delta \to 0} \frac{c^T(f(x + (t+\delta)(y-x)) - c^T(f(x+t(y-x))}{|\delta|} = c^T \delta_{(y-x)} f(x + t(y-x)) \tag{24}$$

Further, there exists a lebesgue integrable function $g(t)$ that equals $h'_c(t)$ almost everywhere and

$$|h(0) - h(1)| = \left| \int_0^1 g(t) d\mu \right| \tag{25}$$

We can assume without loss of generality that

$$g(t) = \begin{cases} h'_c(t) & \text{if } h'_c(t) \text{ exists} \\ \sup_{s \in [0,1]} |h'(s)| & \text{otherwise} \end{cases} \tag{26}$$

where the supremum is defined over all points where $h'_c(t)$ is defined. Then because $g$ agrees almost everywhere with $h'_c$ and is bounded pointwise, we have the following chain of inequalities:

$$||f(x) - f(y)||_\beta = \sup_{||c||_{\beta^*} \leq 1} |h_c(0) - h_c(1)| = \sup_{||c||_{\beta^*} \leq 1} \left| \int_0^1 |g(t) d\mu| \right| \tag{27}$$

$$\leq \sup_{||c||_{\beta^*} \leq 1} \int_0^1 |g(t)| d\mu \tag{28}$$

$$\leq \sup_{||c||_{\beta^*} \leq 1} \int_0^1 \sup_{s \in [0,1]} |h'_c(s)| d\mu \tag{29}$$

$$\leq \sup_{||c||_{\beta^*} \leq 1} \int_0^1 \sup_{s \in [0,1]} |c^T \delta_{(y-x)} f(x + s(y-x))| d\mu \tag{30}$$

$$\leq \sup_{||c||_{\beta^*} \leq 1} \int_0^1 \sup_{z \in \mathcal{D}(y-x)} |c^T \delta_{(y-x)} f(z)| d\mu \tag{31}$$

$$\leq \sup_{||c||_{\beta^*} \leq 1} \int_0^1 ||c||_{\beta^*} \sup_{z \in \mathcal{D}(y-x)} ||\delta_{(y-x)} f(z)||_\beta d\mu \tag{32}$$

$$\leq \sup_{z \in \text{Diff}(\mathcal{X})} ||\nabla f(z)(y-x)||_\beta \tag{33}$$

$$\leq \sup_{z \in \text{Diff}(\mathcal{X})} ||\nabla f(z)||_{\alpha,\beta} ||x - y||_\alpha \tag{34}$$

Where Equation 32 holds by Proposition A.1, Equation 33 holds by Lemma A.1, and the final inequality holds by Proposition A.2. Dividing by $||x - y||_\alpha$ yields the desired result.

On the other hand, we wish to show, for every $\epsilon > 0$, the existence of an $x, y \in \mathcal{X}$ such that

$$\frac{||f(x) - f(y)||_\beta}{||x - y||_\alpha} \geq \sup_{x \in \text{Diff}(\mathcal{X})} ||\nabla f(x)||_{\alpha, \beta} - \epsilon \tag{35}$$

Fix $\epsilon > 0$ and consider any point $z \in \mathcal{X}$ with $||\nabla f(z)^T||_{\alpha, \beta} \geq \sup_{x \in \mathcal{X}} ||\nabla f(x)^T||_{\alpha, \beta} - \epsilon/2$.

Then $||\nabla f(z)^T||_{\alpha, \beta} = \sup_{||v||_\alpha \leq 1} ||\nabla f(z)^T v||_\beta = \sup_{||v||_\alpha \leq 1} ||d_v f(z)||_\beta$. By the definition of the directional derivative, there exists some $\delta > 0$ such that for all $|t| < \delta$,

$$\frac{||f(z + tv) - f(z)||_\beta}{||tv||_\alpha} \geq ||d_v f(z)||_\beta - \epsilon/2 \geq \sup_{x \in \text{Diff}(\mathcal{X})} ||\nabla f(x)^T||_{\alpha, \beta} - \epsilon \tag{36}$$

Hence setting $x = z + tv$ and $y = v$, we recover equation 35. $\qquad\square$

## B. Chain Rule and General Position proofs

In this section, we provide the formal proofs of statements made in Section 3.

### B.1. Preliminaries

**Polytopes**: We use the term polytope to refer to subsets of $\mathbb{R}^d$ of the form $\{x \mid Ax \leq b\}$. The affine hull of a polytope is the smallest affine subspace which contains it. The dimension of a polytope is the dimension of its affine hull. The relative interior of a polytope $P$ is the interior of $P$ within the affine hull of $P$ (i.e., lower-dimensional polytopes have empty interior, but not nonempty relative interior unless the polytope has dimension 0).

**Hyperplanes**: A hyperplane is an affine subspace of $\mathbb{R}^d$ of codimension 1. A hyperplane may equivalently be viewed as the zero-locus of an affine function: $H := \{x \mid a^T x = b\}$. A hyperplane partitions $\mathbb{R}^d$ into two closed halfspaces, $H^+, H^-$ defined by $H^+ := \{x \mid a^T x \geq b\}$ and similarly for $H^-$. When the inequality is strict, we define the open halfspaces as $H_o^+, H_o^-$. We remark that if $U$ is an affine subspace of $\mathbb{R}^d$ and $H$ is a hyperplane that does not contain $U$, then $H \cap U$ is a subspace of codimension 1 relative to $U$. If this is the case, then $H \cap U$ is a subspace of dimension $dim(U) - 1$. A hyperplane $H$ is called a separating hyperplane of a convex set $C$ if $H \cap C = \emptyset$. $H$ is called a supporting hyperplane of $C$ if $H \cap C \neq \emptyset$ and $C$ is contained in either $H^+$ or $H^-$.

**ReLU Kernels**: For a ReLU network, define the functions $g_i(x)$ as the input to the $i^{th}$ ReLU of $f$. We define the $i^{th}$ ReLU kernel as the set for which $g_i = 0$:

$$K_i := \{x \mid g_i(x) = 0\} \tag{37}$$

**The Chain Rule:** The chain rule is a means to compute derivatives of compositions of smooth functions. Backpropagation is a dynamic-programming algorithm to perform the chain rule, increasing efficiency by memoization. This is most easily viewed as performing a backwards pass over the computation graph, where each node has associated with it a partial derivative of its output with respect to its input. As mentioned in the main paper, the chain rule may perform incorrectly when elements of the composition are nonsmooth, such as the ReLU operator. Indeed, the ReLU $\sigma$ has a derivative which is well defined everywhere except for zero, for which it has a subdifferential of $[0, 1]$.

**Definition B.1.** *Consider any implementation of the chain rule which may arbitrarily assign any element of the generalized gradient $\delta_\sigma(0)$ for each required partial derivative $\sigma'(0)$. We define the set-valued function $\nabla^\# f(\cdot)$ as the collection of answers yielded by **any** such chain rule.*

While we note that our mixed-integer programming formulation treats $\nabla^\#(f)$ in this set-valued sense, most implementations of automatic differentiation choose either $\{0, 1\}$ to be the evaluation of $\sigma'(0)$ such that $\nabla^\# f$ is not set valued (e.g.,in PyTorch and Tensorflow, $\sigma'(0) = 0$). Our theory holds for our set-valued formulation, but in the case of automatic differentiation packages, as long as $\sigma'(0) \in [0, 1]$, our results will hold.

**A Remark on Hyperplane Arrangements**: As noted in the main paper, our definition of general position neural networks is spiritually similar to the notion of general position hyperplane arrangements. A hyperplane arrangement $\mathcal{A} := \{H_1, \ldots, H_n\}$ is a collection of hyperplanes in $\mathbb{R}^d$ and is said to be in general position if the intersection of any $k$ hyperplanes is a $(d - k)$ dimensional subspace. Further, if a ReLU network only has one hidden layer, each ReLU kernel is a hyperplane. Thus, hyperplane arrangements are a subset of ReLU kernel arrangements.

### B.2. Proof of Theorem 2

Before restating Theorem 2 and the proof, we introduce the following lemmas:

**Lemma B.1.** *Let $\{K_i\}_{i=1}^m$ be the ReLU kernels of a general position neural net, $f$. Then for any $x$ contained in exactly $k$ of them, say WLOG $K_1, \ldots, K_k$, $x$ lies in the relative interior of one of the polyhedral components of $\cap_{i=1}^k K_i$.*

*Proof.* Since $f$ is in general position, $\cap_{i=1}^k K_i$ is a union of $(d - k)$-dimensional polytopes. Let $P$ be one of the polytopes in this union such that $x \in P$. Since $P$ is an $(d - k)$-face in the polyhedral complex induced by $\{K_i\}_{i=1}^m$, each point on the boundary of $P$ is the intersection of at least $k + 1$ ReLU kernels of $f$. Thus $x$ cannot be contained in the boundary of $P$ and must reside in the relative interior. $\square$

The rest of the components are geometric. We introduce the notion of a cutting hyperplane:

*Figure 1.* Examples of the three classes of hyperplanes with respect to a polytope $P$ (pink). The blue hyperplane is a separating hyperplane of $P$, the green hyperplane is a supporting hyperplane of $P$, and the red hyperplane is a cutting hyperplane of $P$.

**Definition B.2.** *We say that a hyperplane $H$ is a **cutting hyperplane** of a polytope $P$ if it is neither a separating nor supporting hyperplane of $P$.*

We now state and prove several properties of cutting hyperplanes:

**Lemma B.2.** *The following are equivalent:*

(a) *$H$ is a cutting hyperplane of $P$.*

(b) *$H$ contains a point in the relative interior of $P$, and $H \cap P \neq P$.*

(c) *$H$ cuts $P$ into two polytopes with the same dimension as $P$: $dim(P \cap H^+) = dim(P \cap H^-) = dim(P)$ and $H \cap P \neq P$.*

*Proof.* Throughout we will denote the affine hull of $P$ as $U$.

(a) $\implies$ (b): By assumption, $H$ is neither a supporting nor separating hyperplane. Since neither $H^+ \cap P$ nor $H^- \cap P$ is $P$, $H \cap P \neq P$. Thus $H \cap P$ is a codimension 1 subspace, with respect to $U$. Since $H$ is not a supporting hyperplane, $H \cap U$ must not lie on the boundary of $P$ (relative to $U$). Thus $H \cap U$ contains a point in the relative interior of $P$ and so does $H$.

(b) $\implies$ (c): By assumption $H \cap P \neq P$. Consider some point, $x$, inside $H$ and the relative interior of $P$. By definition of relative interior, there is some neighborhood $N_\epsilon(x)$ such that $N_\epsilon(x) \cap U \subset P$. Thus there exists some $x' \in N_\epsilon(x)$ such that $(N_{\epsilon'}(x') \cap U) \subset (H_o^+ \cap P)$ and thus the affine hull of $H^+ \cap P$ must have the same dimension as $U$. Similarly for $H^- \cap P$.

(c) $\implies$ (a): Since $P \cap H^+$ and $P \cap H^-$ are nonempty, then $P \cap H$ is nonempty and thus $H$ is not a separating hyperplane of $P$. Suppose for the sake of contradiction that $H^+ \cap P = P$. Then $H_o^- \cap P = \emptyset$, this implies that $dim(H \cap P) = dim(P)$ which only occurs if $P \subseteq H$ which is a contradiction. Repeating this for $H^- \cap P$, we see that $H$ is not a supporting hyperplane of $P$. $\qquad\square$

**Lemma B.3.** *Let $F$ be a $(k)$-dimensional face of a polytope $P$. If $H$ is a cutting hyperplane of $F$, then $H$ is a cutting hyperplane of $P$.*

*Proof.* Since $H$ is a cutting hyperplane of $F$, $H$ is neither a separating hyperplane nor is $P \subseteq H$. Thus it suffices to show that $H$ is not a supporting hyperplane of $P$. Since $H$ cuts $F$, there exist points inside $F \cap H_o^+$ and $F \cap H_o^-$, where $H_o^+$, $H_o^-$ are the open halfspaces induced by $H$. Thus neither $P \cap H_o^+$ nor $P \cap H_o^-$ are empty, which implies that $H$ is not a supporting hyperplane of $P$, hence $H$ must also be a cutting hyperplane of $P$.

$\qquad\square$

Now we can proceed with the proof of Theorem 2:

**Theorem B.1.** *Let $f$ be a general position ReLU network, then for every $x$ in the domain of $f$, the set of elements returned by the generalized chain rule $\nabla^{\#} f(x)$ is exactly the generalized Jacobian:*

$$\nabla^{\#} f(x) = \delta_f(x) \tag{38}$$

*Proof.* **Part 1:** The first part of this proof shows that if $x$ is contained in exactly $k$ ReLU kernels, then $x$ is contained in $2^k$ full-dimensional linear regions of $f$. We prove this claim by induction on $k$. The case where $k = 1$ is trivial. Now assume that the claim holds up to $k - 1$. Assume that $x$ lies in the ReLU kernel for every neuron in a set $S \subseteq [m]$, with $|S| = k$. Without loss of generality, let $j \in S$ be a neuron whose depth, $L$, is at least as great as the depth of every other neuron in $S$. Then one can construct a subnetwork $f'$ of $f$ by considering only the first $L$ layers of $f$ and omitting neuron $j$. Now $K_i$ is a ReLU kernel of $f'$ for every $i \in S \setminus \{j\}$, and further suppose that $f'$ is a general position ReLU net. From the inductive hypothesis, we can see that $x$ is contained in exactly $2^{k-1}$ linear regions of $f'$. By Lemma B.1, $x$ resides in the relative interior of a $(n - k + 1)$-dimensional polytope, $P$, contained in the union that defines $\cap_{i=1}^{k-1} K_i$. Since $j$ has maximal depth, $g_j(\cdot)$ is affine in $P$, and thus there exists some hyperplane $H$ such that $P \cap K_j = P \cap H$. Thus by Lemma B.2 (b), $H$ is a cutting hyperplane of $P$.

Consider some linear region $R$ of $f'$ containing $x$. Then $g_j(x)$ is affine inside each $R$ and hence there exists some hyperplane $H_R$ such that $R \cap K_j = R \cap H_R$, with the additional property that $H_R \cap P = H \cap P$. By general position, $H \cap P \neq P$ and thus $H_R$ is a cutting hyperplane for $P$ by Lemma B.2 (b). Since $P$ is a $(n - k + 1)$-dimensional face of $R$, we can apply Lemma B.3 to see that $H_R$ is a cutting hyperplane for $R$ as desired.

**Part 2:**

Now we show that the implication proved in part 1 of the proof implies that $\nabla^{\#} f(x) = \delta_f(x)$. This follows in two steps. The first step is to show that $\nabla f^{\#}(x)$ is a convex set for all $x$, and the second step is to show the following inclusion holds:

$$\mathcal{V}(\delta_f(x)) = \mathcal{V}(\nabla^{\#} f(x)) \tag{39}$$

Where, for any convex set $C$, $\mathcal{V}(C)$ denotes the set of extreme points of $C$. Then the theorem will follow by taking convex hulls.

To show that $\nabla^{\#} f(x)$ is convex, we make the following observation: every element of $\nabla^{\#} f(x)$ must be attainable by some implementation of the chain rule which assigns values for every $\sigma'(0)$. If $\Lambda_0 \in \nabla^{\#} f(x)$ is attainable by setting exactly zero $\sigma'(0)'s$ to lie in the open interval $(0, 1)$, then $\Lambda_0$ is the Jacobian matrix corresponding to one of the full-dimensional linear regions that $x$ is contained in. Consider some $\Lambda_r \in \nabla^{\#} f(x)$ which is attainable by setting exactly $r$ $\sigma'(0)'s$ to lie in the open interval $(0, 1)$. Then certainly $\Lambda_r$ may be written as the convex combination of $\Lambda_{r-1}^{(1)}$ and $\Lambda_{r-1}^{(2)}$ for two elements of $\delta_f(x)$, attainable by setting exactly $(r - 1)$ ReLU partial derivatives to be nonintegral. This holds for all $r \in \{1, \dots k\}$ and thus $\nabla^{\#} f(x)$ is convex.

To show the equality in Equation 39, we first consider some element of $\mathcal{V}(\delta_f(x))$. Certainly this must be the Jacobian of some full-dimensional linear region containing $x$, and hence there exists some assignment of ReLU partial derivatives such that the chain rule yields this Jacobian. On the other hand, we've shown in the previous section that every element of $\nabla^{\#} f(x)$ may be written as a convex combination of the Jacobians of the full-dimensional linear regions of $f$ containing $x$. Hence each extreme point of $\nabla^{\#} f(x)$ must be the Jacobian of one of the full-dimensional linear regions of $f$ containing $x$. □

## B.3. Proof of Theorem 3

Before presenting the proof of Theorem 3, we will more explicitly define a Lebesgue measure over parameter space of a ReLU network. Indeed, consider every ReLU network with a fixed architecture and hence a fixed number of parameters. We can identify each of these parameters with $\mathbb{R}$ such that the parameter space of a ReLU network with $k$ parameters is identifiable with $\mathbb{R}^k$. We introduce the measure $\mu_f$ as the Lebesgue measure over neural networks with the same architecture as a defined ReLU network $f$. Now we present our Theorem:

**Theorem B.2.** *The set of ReLU networks not in general position has Lebesgue measure zero over the parameter space.*

*Proof.* We prove the claim by induction over the number of neurons of a ReLU network. As every ReLU network with only one neuron is in general position, the base case holds trivially. Now suppose that the claim holds for families of ReLU

networks with $k-1$ neurons. Then we can add a new neuron in one of two ways: either we add a new neuron to the final layer, or we add a new layer with only a single neuron. Every neural network may be constructed in this fashion, so the induction suffices to prove the claim. Both cases of the induction may be proved with the same argument:

Consider some ReLU network, $f$, with $k-1$ neurons. Then consider adding a new neuron to $f$ in either of the two ways described above. Let $B_f$ denote the set of neural networks with the same architecture as $f$ that are not in general position, and similarly for $B_{f'}$. Let $C_{f'}$ denote the set of neural networks with the same architecture as $f'$ that are not in general position, but are in general position when the $k^{th}$ neuron is removed. Certainly if $f$ is not in general position, then $f'$ is not in general position. Thus

$$\mu_{f'}(B'_f) \leq \mu_f(B_f) + \mu_{f'}(C_{f'}) = \mu_{f'}(C_{f'}) \tag{40}$$

where $\mu_f(B_f) = 0$ by the induction hypothesis. We need to show the measure of $C_{f'}$ is zero as follows. Letting $K_k$ denote the ReLU kernel of the neuron added to $f$ to yield $f'$, we note that $f$ is not in general position only if one of the affine hulls of the polyhedral components of $K_k$ contains the affine hull of some polyhedral component of some intersection $\cap_{i \in S} K_i$ where $S$ is a nonempty subset of the $k-1$ neurons of $f$. We primarily control the bias parameter, as this is universal over all linear regions, and notice that this problem reduces to the following: what is the measure of hyperplanes that contain any of a finite collection of affine subspaces? By the countable subadditivity of the Lebesgue measure and the fact that the set of hyperplanes that contain any single affine subspace has measure 0, $\mu_{f'}(C_{f'}) = 0$. $\qquad \square$

## C. Complexity Results

### C.1. Complexity Theory Preliminaries

Here we recall some relevant preliminaries in complexity theory. We will gloss over some formalisms where we can, though a more formal discussion can be found here (Williamson and Shmoys, 2011; Hromkovič, 2013; Demaine, 2014).

We are typically interested in combinatorial optimization problems, which we will define informally as follows:

**Definition C.1.** *A **combinatorial optimization problem** is composed of 4 elements: i) A set of valid **instances**; ii) A set of feasible **solutions** for each valid instance; iii) A non-negative **cost** or **objective value** for each feasible solution; iv) A **goal**: signifying whether we want to find a feasible solution that either minimizes or maximizes the cost function.*

In this subsection, we will typically refer to problems using the letter $\Pi$, where instances of that optimization problem are $x$, and feasible solutions are $y$, and the cost of $y$ is $m(y)$. We will refer to the **cost** of the optimal solution to instance $x \in \Pi$ as $OPT(x)$. Optimization problems then typically have 3 formulations, listed in order of decreasing difficulty:

- **Search Problem:** Given an instance $x$ of optimization problem $\Pi$, find $y$ such that $m(y) = OPT(x)$.

- **Computational Problem**: Given an instance $x$ of optimization problem $\Pi$, find $OPT(x)$

- **Decision Problem**: Given an instance $x$ of optimization problem $\Pi$, and a number $k$, decide whether or not $OPT(x) \geq k$.

Certainly an efficient algorithm to do one of these implies an efficient algorithm to do the next one. Also note that by a binary search procedure, the computational problem is polynomially-time reducible to the decision problem. As complexity theory is typically couched in discussion about membership in a language, it is slightly awkward to discuss hardness of combinatorial optimization problems. Since, every computational flavor of an optimization problem has a poly-time equivalent decision problem, we will simply claim that an optimization problem is NP-hard if its decision problem is NP-hard.

While many interesting optimization problems are hard to solve exactly, for many of these interesting problems there exist efficient approximation algorithms that can provide a guarantee about the cost of the optimal solution.

**Definition C.2.** *For a maximization problem $\Pi$, an approximation algorithm with approximation ratio $\alpha$ is a polynomial-time algorithm that, for every instance $x \in \Pi$, produces a feasible solution, $y$, such that $m(y) \geq OPT(x)/\alpha$.*

Noting that $\alpha > 1$ can either be a constant or a function parameterized by $|x|$, length of the binary encoding of instance $x$. We also note that this definition frames approximation algorithms as a "search problem".

A very powerful tool in showing the hardness of approximation problems is the notion of a $c$-gap problem. This is a form of promise problem, and proofs of hardness here are slightly stronger than what we actually desire.

**Definition C.3.** *Given an instance of an maximization problem $x \in \Pi$ and a number $k$, the **c-gap problem** aims to distinguish between the following two cases:*

- YES: $OPT(x) \geq k$

- NO: $OPT(x) < k/c$

*where there is no requirement on what the output should be, should $OPT(x)$ fall somewhere in $[k/c, k)$. For minimization problems, YES cases imply $OPT(x) \leq k$, and NO cases imply $OPT(x) > k \cdot c$.*

Again we note that $c$ may be a function that takes the length of $x$ as an input. We now recall how a $c$-approximation algorithm may be used to solve the $c$-gap problem, implying the $c$-gap problem is at least as hard as the $c$-approximation.

**Proposition C.1.** *If the c-gap problem is hard for a maximization problem $\Pi$, then the c-approximation problem is hard for $\Pi$.*

*Proof.* Suppose we have an efficient $c$-approximation algorithm for $\Pi$, implying that for any instance $x \in \Pi$, we can output a feasible solution $y$ such that $OPT(x)/c \leq m(y) \leq OPT(x)$. Then we let $A_k$ be an algorithm that retuns YES if

$m(y) \geq k/c$, and NO otherwise, where $y$ is the solution returned by the approximation algorithm Then for the gap-problem, if $OPT(x) \geq k$, we have that $m(y) \geq k/c$ so the $A_k$ will output YES. On the other hand, if $OPT(x) < k/c$, then $A_k$ will output NO. Hence, $A_k$ is an efficient algorithm to decide the $c$-gap problem. □

While hardness of approximation results arise from various forms, most notably the PCP theorem, we can black-box the heavy machinery and prove our desired results using only strict reductions, which we define as follows.

**Definition C.4.** *A **strict reduction** from problem $\Pi$ to problem $\Pi'$, is a functions $f$, such that $f : \Pi \to \Pi'$ maps problem instances of $\Pi$ to problem instances of $\Pi'$. $f$ must satisfy the following properties that for all $x \in \Pi$*

1. *$\frac{|f(x)|}{|x|} \leq \alpha$, where $\alpha$ is a fixed constant*

2. *$OPT_\Pi(x) = OPT_{\Pi'}(f(x))$*

For which we can now state and prove the following useful proposition:

**Proposition C.2.** *If $f, g$ are a strict reduction from optimization problem $\Pi$ to optimization problem $\Pi'$, and the $c$-gap problem is hard for $\Pi$, where $c$ is polynomial in the size of $|x|$, then the $c'$-gap problem is hard for $\Pi'$, where $c' \in \Theta(c)$.*

*Proof.* Suppose both $\Pi$ and $\Pi'$ are maximization problems, and the $c$-gap problem is hard for $\Pi$. We consider the case where $c$ is a function that takes as input the encoding size of instances of $\Pi$. We can define the function $c'(n) := c(n/\alpha)$ for all $n$. Hence $c(|x|) = c'(|f(x)|)$ for all $x \in \Pi$ by point 1 of the definition of strict reduction. Then for all $k$ and all $x \in \Pi$, the following two implications hold

$$OPT_{\Pi'}(f(x)) \geq k \implies OPT_\Pi(x) \geq k$$

$$OPT_{\Pi'}(f(x)) < \frac{k}{c'(f|x|)} \implies OPT_\Pi(x) < \frac{k}{c(|x|)}$$

Where both implications hold because $OPT_\Pi(x) = OPT_{\Pi'}(f(x))$. If the $c'$-gap problem were efficiently decidable for $\Pi'$, then the $c'$-gap problem would be efficiently decidable for $\Pi$.

If $\Pi$ is a maximization problem and $\Pi'$ is a minimization problem, then the following two implications hold:

$$OPT_{\Pi'}(f(x)) \leq k' \implies OPT_\Pi(x) \leq k'$$

$$OPT_{\Pi'}(f(x)) > k' \cdot c'(|f(x)|) \implies OPT_\Pi(x) > k' \cdot c(|x|)$$

Then letting $k = k' \cdot c(|x|)$ we have that solving the $c'$-gap problem for $\Pi'$ would solve the $c$-gap problem for $\Pi$. The proofs for $\Pi, \Pi'$ both being minimization problems, or $\Pi$ being a minimization and $\Pi'$ being a maximization hold using similar strategies. □

### C.2. Proof of Theorem 4

Now we return to ReLU networks and prove novel results about the inapproximability of computing the local Lipschitz constant of a ReLU network. Recall that we have defined ReLU networks as compositions of functions of the form :

$$f(x) = c^T \sigma\left(Z_d(x)\right) \qquad\qquad Z_i(x) = W_i \sigma\left(Z_{i-1}(x)\right) + b_i, \qquad\qquad (41)$$

where $Z_0(x) = x$ and $\sigma$ is the elementwise ReLU operator. In this section, we only consider scalar-valued general position ReLU networks, $f : \mathbb{R}^n \to \mathbb{R}$. Towards the end of the proof we see that we show that the general position assumption is not needed for our construction. We can formulate the chain rule as follows:

**Proposition C.3.** *If $x$ is contained in the interior of some linear region of a general position ReLU network $f$, then the chain rule provides the correct gradient of $f$ at $x$, where the $i^{th}$ coordinate of $\nabla f(x)$ is given by:*

$$\nabla f(x)_i = \sum_{\Gamma \in Paths(i)} \left( \prod_{w_j \in \Gamma} w_j \right)$$

*where $Paths(i)$ is the set of paths from the $i^{th}$ input, $x_i$, to the output in the computation graph, where the ReLU at each vertex is on, and $w_j$ is the weight of the $j^{th}$ edge along the path.*

We define the following optimization problems:

**Definition C.5.** `MAX-GRAD` *is an optimization problem, where the set of valid instances is the set of scalar-valued ReLU networks. The feasible solutions are the set of differentiable points $x \in \mathcal{X}$, which have cost $||\nabla f(x)||_1$. The goal is to maximize this gradient norm.*

**Definition C.6.** `MIN-LIP` *is an optimization problem where the set of valid instances is the set of piecewise linear neural nets. The feasible solutions are the set of constants $L$ such that $L \geq L(f)$. The cost is the identity function, and our goal is to minimize $L$.*

Of course, each of these problems have decision-problem variants, denoted by `MAX-GRAD`$_{dec}$ and `MIN-LIP`$_{dec}$. We also remark that by Theorem A.1, and proposition C.2, the trivial strict reduction implies that it is at least as hard to approximate `MIN-LIP` as it is to approximate `MAX-GRAD`. For the rest of this section, we will only strive to prove hardness and inapproximability results for `MAX-GRAD`.

To do this, we recall the definition of the maximum independent set problem:

**Definition C.7.** `MIS` *is an optimization problem, where valid instances are undirected graphs $G = (V, E)$, and feasible solutions are $U \subseteq V$ such that for any $v_i, v_j \in U$, $(v_i, v_j) \notin E$. The cost is the size of $U$, and the goal is to maximize this cost.*

Classically, it has been shown that `MIS` is NP-hard to optimize, but also is one of the hardest problems to approximate and does not admit a deterministic polynomial time algorithm to solve the $O(|V|^{1-\epsilon})$-gap problem (Zuckerman, 2007).

For ease of exposition, we rephrase instances of `MIS` into instances of an equivalent problem which aims to maximize the size of consistent collections of locally independent sets. Given graph $G = (V, E)$, for any vertex $v_i \in V$, we let $N(v_i)$ refer to the set of vertices adjacent to $V_i$ in $G$. We sometimes will abuse notation and refer to variables by their indices, e.g., $N(i)$. We also refer to the degree of vertex $i$ as $d(v_i)$ or $d(i)$.

**Definition C.8.** *A **locally indpendent set** centered at $v_i$ is a $\{-1, +1\}$-labelling of the vertices $\{v_i\} \cup N(v_i)$ such that the label of $v_i$ is $+1$ and the label of $v_j \in N(v_i)$ is $-1$. Two locally independent sets are said to be **consistent** if, for every $v_j$ appearing in both locally independent sets, the label is the same in both locally independent sets. A **consistent collection** of locally independent sets is a set of locally independent sets that is pairwise consistent.*

Then we can define an optimization problem:

**Definition C.9.** `LIS` *is an optimization problem, where valid instances are undirected graphs $G = (V, E)$, and feasible solutions are consistent collections of locally indpendent sets. The cost is the size of the collection, and the goal is to maximize this cost.*

It is obvious to see that there is a trivial strict reduction between `MIS` and `LIS`. Indeed, any independent set defines the centers of a consistent collection of locally independent sets, and vice versa. As we will see, this is a more natural problem to encode with neural networks than `MIS`.

Now we can state our first theorem about the inapproximability of `MAX-GRAD`.

**Theorem C.1.** *Let $f$ be a scalar-valued ReLU network, not necessarily in general position, taking inputs in $\mathbb{R}^d$. Then assuming the exponential time hypothesis, there does not exist a polynomial-time approximation algorithm with ratio $\Omega(d^{1-c})$ for computing $L^\infty(f, \mathcal{X})$ and $L^1(f, \mathcal{X})$ for any constant $c > 0$.*

*Proof.* We will prove this first for $L^\infty(f, \mathcal{X})$ and then slightly modify the construction to prove this for $L^1(f, \mathcal{X})$. We will throughout take $\mathcal{X} := \mathbb{R}^d$. The key idea of the proof is to, given a graph $G = (V, E)$ with $|V| =: n$, encode a neural network $h$ with $n$ inputs, each representing the labeling value. We then build a neuron for each possible locally independent set, where the neuron is 'on' if and only if the labelling is close to a locally independent set. And then we also ensure that each locally independent set contributes $+1$ to the norm of the gradient of $f$.

A critical gadget we will use is a function $\psi(x) : \mathbb{R} \to \mathbb{R}$ defined as follows:

which is implementable with affine layers and ReLU's as: $\psi(x) = \sigma(x+1) - \sigma(x-1) - 1$. We are now ready to construct our neural net. For every vertex $v_i$ in $V$, we construct an input to the neural net, hence $f : \mathbb{R}^n \to \mathbb{R}$. We denote the $i^{th}$ input to $f$ as $x_i$. The first order of business is to map each $x_i$ through $\psi(\cdot)$, which can be done by two affine layers and one ReLU layer. e.g., we can define $\psi(x_i) = A_1(ReLU(A_0(x_i)))$ where

$$\psi(x) = \begin{cases} -1 & x \in (-\infty, -1] \\ x & x \in [-1, 1] \\ 1 & x \in [1, +\infty) \end{cases} \tag{42}$$

$$A_0(x) := \begin{bmatrix} 1 \\ 1 \end{bmatrix} x + \begin{bmatrix} 1 \\ -1 \end{bmatrix}$$

$$A_1(z) := \begin{bmatrix} 1 & | & -1 \end{bmatrix} z - 1$$

Next we define the second layer of ReLU's, which has width $n$, and each neuron represents the status of a locally indpendent set. We define the input to the $i^{th}$ ReLU in this layer as $I_i$ with

$$I_i(x) := \psi(x_i) - \sum_{j \in N(i)} \psi(x_j) - (d + 1 - \epsilon) \tag{43}$$

for some fixed-value $\epsilon$ to be chosen later. Finally, we conclude our construction with a final affine layer to our neural net as

$$h(x) := \sum_{i=1}^{n} \frac{\sigma(I_i(x))}{d(i) + 1} \tag{44}$$

Let $\mathcal{I}(x)$ denote the set of indices of ReLU's that are 'on' in the second-hidden layer of $h$: $\mathcal{I}(x) := \{i \mid I_i(x) > 0\}$. Now we make the following claims about the structure of $h$.

**Claim C.1.** *For every $i$, if $i \in \mathcal{I}(x)$ then $x_i > 1 - \epsilon$ and $x_j < -1 + \epsilon$ for all $j \in N(i)$. In addition, $\mathcal{I}(x)$ denotes the centers of a consistent collection of locally independent sets.*

*Proof.* Indeed, if $I_i(x) > 0$, then the sum of $(d(i) + 1)$ $\psi$-terms is greater than $1 - \epsilon$. As each $\psi$-term is in the range $[-1, 1]$, each $\psi$-term must individually be at least $1 - \epsilon$. And $\psi(x_i) \geq 1 - \epsilon$ implies $x_i \geq 1 - \epsilon$. Similarly, $-\psi(x_j) \geq 1 - \epsilon$ implies that $x_j \leq -1 + \epsilon$. Now consider any $i_1, i_2$ in $\mathcal{I}(x)$. Then the pair of locally independent sets centered at $v_{i_1}$ and $v_{i_2}$ is certainly consistent. $\square$

*Figure 2.* Complete construction of a neural network $h$ that is the reduction from `LIS`, such that the supremal gradient of $h$ corresponds to the maximum locally indendent set. The first step is to map each $x_i$ to $\psi(x_i)$, then to construct $I_i(x)$. Finally, we route each $\sigma(I_i(x))$ to the output.

**Claim C.2.** *For any $x$ such that $h$ is differentiable at $x$, $\nabla h(x)_i \cdot x_i \geq 0$.*

*Proof.* We split into cases based on the value of $x_i$ and rely on Claim C.1. Suppose $x_i \in (-1 + \epsilon, 1 - \epsilon)$, then we have $I_j < 0$ for any $j$ in $\{i\} \cup N(i)$ and hence $\nabla h(x)_i = 0$. If $x_i \geq 1 - \epsilon$, then every $j \in N(i)$ has $I_j < 0$ and hence by Proposition C.3, the only contributions to the $\nabla h(x)_i$ can be from paths that route from $x_i$ to the output through $I_i$. Hence

$$\nabla h(x)_i = \frac{\delta h}{\delta I_i} \cdot \frac{\delta I_i}{\delta x_i}, \tag{45}$$

where both terms are nonnegative and hence so is $\nabla h(x)_i$. Finally, if $x_i \leq -1 + \epsilon$, then the only contributions to $\nabla h(x)_i$ come from paths that route through $I_j$ for $j \in N(i)$, hence

$$\nabla h(x)_i = \sum_{j \in N(i)} \frac{\delta h}{\delta I_j} \cdot \frac{\delta I_j}{\delta x_i} \tag{46}$$

where the first term is nonnegative and the second term is always nonpositive. We remark that because we have assumed $h$ to be differentiable at $x$, the chain rule provides correct answers, by Rademacher's theorem. $\qquad\square$

**Claim C.3.** *For any $x$, let $\mathcal{I}(x)$ be defined as above, $\mathcal{I}(x) := \{i \mid I_i(x) > 0\}$. Then $||\nabla h(x)||_1 \leq |\mathcal{I}(x)|$ and for every $x$. In addition, for every $x$, there exists a $y$ with $||\nabla h(y)||_1 \geq |\mathcal{I}(x)|$.*

*Proof.* To show the first part, observe that

$$h(x) = \sum_{i=1}^{n} \frac{\sigma\left(I_i(x)\right)}{d(i) + 1} = \sum_{i \in \mathcal{I}(x)} \frac{I_i(x)}{d(i) + 1} \implies \nabla h(x) = \sum_{i \in \mathcal{I}(x)} \frac{\nabla I_i(x)}{d(i) + 1} \tag{47}$$

hence

$$||\nabla h(x)||_1 \leq \sum_{i \in \mathcal{I}(x)}^{n} \frac{1}{d(i) + 1} ||\nabla I_i(x)||_1 \tag{48}$$

and

$$I_i(x) := \psi(x_i) - \sum_{j \in N(i)} \psi(x_j) - (d + 1 - \epsilon) \implies \nabla I_i(x) = \nabla \psi(x_i) - \sum_{j \in N(i)} \nabla \psi(x_j) \tag{49}$$

hence

$$||\nabla I_i(x)||_1 \leq ||\nabla \psi(x_i)||_1 + \sum_{j \in N(i)} ||\nabla \psi(x_j)||_1 \leq d(i) + 1 \tag{50}$$

where the final inequality follows because $||\nabla \psi(x_i)||_1 \leq 1$ everywhere it is defined. Combining equations 48 and 50 yields that $||\nabla h(x)||_1 \leq |\mathcal{I}(x)|$.

On the other hand, suppose $\mathcal{I}(x)$ is given. Then we can construct $y$ such that $\mathcal{I}(y) = \mathcal{I}(x)$ and $||\nabla h(y)||_1 = |\mathcal{I}(x)|$. To do this, set $y_i = 1 - \frac{\epsilon}{2n}$ if $i \in \mathcal{I}(x)$ and $y_i = -1 + \frac{\epsilon}{2n}$ otherwise. Then note that $\mathcal{I}(x) = \mathcal{I}(y)$ and for every $i \in \mathcal{I}(y)$

$$\nabla I_i(y)_k = \begin{cases} +1 & k = i \\ -1 & k \in N(i) \\ 0 & \text{otherwise} \end{cases}$$

and hence by Claim 2, we can replace the inequalities in equations 48 and 50 we have that

$$||\nabla h(y)||_1 = \sum_{i \in \mathcal{I}(x)} \frac{1}{d(i) + 1} ||\nabla I_i(y)||_1 = |\mathcal{I}(x)|$$

as desired.

$\qquad\square$

To demonstrate that this is indeed a strict reduction, we need to define functions $f$ and $g$, where $f$ maps instances of `LIS` to instances of `MAX-GRAD`, and $g$ maps feasible solutions of `MAX-GRAD` back to `LIS`. Clearly the construction we have defined above is $f$. The function $g$ can be attained by reading off the indices in $\mathcal{I}(x)$.

To demonstrate that the size of this construction does not blow up by more than a constant factor, observe that by representing weights as sparse matrices, the number of nonzero weights is a constant factor times the number of edges in $G$. Indeed, encoding each $\psi$ in the first layer takes $O(1)$ parameters for each vertex in $G$. Encoding $I_i(x)$ requires only $O(d(i))$ parameters for each $i$, and hence $2|E|$ parameters total. Assuming a RAM model where numbers can be represented by single atomic units, and $\epsilon$ is chosen to be $(n+2)^{-1}$, this is only a constant factor expansion.

The crux of this argument is to demonstrate that $OPT_{\text{MAX-GRAD}}(f(q)) = OPT_{\text{LIS}}(q)$. It suffices to show that for every locally independent set $L$, there exists an $x$ such that $||\nabla h(x)||_1 \geq k$, and for every $y$, there exists a locally independent set $L'$ such that $|L'| \geq ||\nabla h(y)||_1$.

Suppose $L$ is a consistent collection of locally independent sets, with $|L| = k$. Any consistent collection of locally independent sets equivalently defines a labelling of each vertex of $G$, where $l_i$ denotes the label of vertex $v_i$: $l_i := +1$ if the locally independent set centered at $v_i$ is contained in the collection, and $l_i := -1$ otherwise. Then one can construct an $x$ such that $||\nabla h(x)||_1 \geq k$. Indeed, for every $v_i$ with label $l_i$, set $x_i = l_i(1 - \frac{\epsilon}{2n})$. By Claim C.1, under this $x$, $\mathcal{I}_i(x) \geq 0$ for every $i$ such that $l_i = +1$, $I_i(x) > 0$. Then $|\mathcal{I}(x)| = k$ and by Claim C.3 there exists a $y$ such that $||\nabla h(y)||_1 \geq k$.

On the other hand, suppose the maximum gradient of $h$ is $k$. Then there exists an $x$ that attains this and by Claim C.3, $|\mathcal{I}(x)| \geq k$. By Claim C.1, we have that $\mathcal{I}(x)$ denotes the centers of a consistent collection of locally independent sets.

**Approximating the maximum $\ell_\infty$ norm:** We only slightly modify our construction of the reduction for `MAX-GRAD` to show inapproximability of $L^1(f, \mathcal{X})$. Namely, we add a new input so $h : \mathbb{R}^{n+1} \to \mathbb{R}$, called $x_{n+1}$. Then we map $x_{n+1}$ through $\psi$ like all the other indices, but instead redefine

$$I_i(x) := \psi(x_i) + (d(i) + 1)\psi(x_{n+1}) - \sum_{j \in N(i)} \psi(x_j) - 2(d(i) + 1 - \epsilon)$$

$$H(x) := \sum_{i=1}^{n} \frac{\sigma(I_i(x))}{2(d(i) + 1)}$$

The rest of this contsruction is nearly identical to the preceding construction, with the exception being that $||\nabla h(x)||_1$ can be replaced by $\frac{\delta h(x)}{\delta x_{n+1}}$ throughout as an indicator to count the size of $\mathcal{I}(x)$.

**About General Position:** Finally we note that this proof is valid even when you do not assume the network be in general position. Observe that the optimal value of the gradient norm is attained at a point in the strict interior of some linear region. Next observe that a network not being in general position may only increase the optimal objective value over what is reported by the Jacobians of the linear regions of $f$, but this cannot happen by claim C.3. Thus general position-ness has no bearing on this construction.

$\square$

As an aside, we note that the strict reduction demonstrates that `MAX-GRAD`$_{dec}$ is NP-complete, which implies that `MIN-LIP`$_{dec}$ is CoNP-complete.

# D. LipMIP Construction

In this appendix we will describe in detail the necessary steps for LipMIP construction. In particular, we will present how to formulate the gradient norm $||\nabla^{\#} f||_{*}$ for scalar-valued, general position ReLU network $f$ as a composition of affine, conditional and switch operators. Then we will present the proofs of MIP-encodability of each of these operators. Finally, we will describe how the global upper and lower bounds are obtained using abstract interpretation.

## D.1. MIP-encodable components of ReLU networks

Our aim in this section is to demonstrate how $||\nabla^{\#} f||_{*}$ may be written as a composition of affine, conditional, and switch operators. For completeness, we redefine these operators here:

**Affine operators** $A : \mathbb{R}^n \to \mathbb{R}^m$ are defined as

$$A(x) = Wx + b, \tag{51}$$

for some fixed matrix $W$ and vector $b$.

The **conditional operator** $C : \mathbb{R} \to \mathcal{P}(\{0, 1\})$ is defined as

$$C(x) = \begin{cases} \{1\} & \text{if } x > 0 \\ \{0\} & \text{if } x < 0 \\ \{0, 1\} & \text{if } x = 0 \end{cases} \tag{52}$$

The **switch operator** $S : \mathbb{R} \times \{0, 1\} \to \mathbb{R}$ is defined as

$$S(x, a) = x \cdot a. \tag{53}$$

We will often abuse notation, and let conditional and switch operators apply to vectors, where the operator is applied elementwise. Now we recover Lemma 1 from section 5 of the main paper.

**Lemma D.1.** *Let $f$ be a scalar-valued general position ReLU network. Then $f(x)$, $\nabla^{\#} f(x)$, $|| \cdot ||_1$, and $|| \cdot ||_\infty$ may all be written as a composition of affine, conditional and switch operators.*

*Proof.* We recall that $f$ is defined recursively like:

$$f(x) = c^T \sigma \left( Z_d(x) \right) \qquad Z_i(x) = W_i \sigma \left( Z_{i-1}(x) \right) + b_i \qquad Z_0(x) = x \tag{54}$$

It amounts to demonstrate how $Z_i(x)$ may be computed as a composition of affine, conditional and switch operator. Since $\sigma(x) = S(x, C(x))$, one can write, $Z_1(x) = W_i S(x, C(x)) + b_i$. Letting $\Lambda_i(x) := C(Z_i(x))$ and $A_i(x) := W_i(x) + b_i$, one can write $Z_i(x) = A_i \circ S \left( Z_i(x), \Lambda_i(x) \right)$. Since $f(x)$ is an affine operator applied to $Z_d(x)$, $f(x)$ can certainly be encoded using only affine, switch, and conditional operators.

To demonstrate that $\nabla f(x)$ may also be written as such a composition, we require the same definition to compute $Z_i(x)$ as above. Then by the chain rule, we have that

$$\nabla^{\#} f(x) = W_1^T Y_1(x) \qquad Y_i(x) = W_{i+1}^T \text{Diag}(\Lambda_i(x)) Y_{i+1}(x) \qquad Y_{d+1}(x) = c \tag{55}$$

As the $\nabla^{\#} f(x)$ is an affine operator applied to $Y_1(x)$, and $Y_{d+1}(x)$ is constant, we only need to show that $Y_i(x)$ may be written as a composition of affine, conditional, and switch operators. This follows from the fact that

$$\text{Diag}(\Lambda_i(x)) Y_{i+1}(x) = S(Y_{i+1}(x), \Lambda_i(x)) \tag{56}$$

Then letting $A_i^T(x) := W_i^T x$ we have that $Y_i(x) = A_i^T \circ (S \left( Y_{i+1}(x), \Lambda_i(x) \right)$. Hence $\nabla^{\#} f(x)$ may be encoded as a composition of affine, conditional, and switch operators.

All that is left is to show that $|| \cdot ||_1, || \cdot ||_\infty$ may be encoded likewise. For each of these, we require $| \cdot |$ which can equivalently be written $|x| = \sigma(x) + \sigma(-x)$, and hence $|x| = S(x, C(x)) + S(-x, C(-x))$. $||x||_1$ then is encoded as the sum of the elementwise sum over $|x_i|$. $|| \cdot ||_\infty$ requires the $\max(\dots)$ operator. To encode this, we see that $\max(x_1, \dots) = \max(x_1, \max(\dots))$ and $\max(x, y) = x + \sigma(y - x) = x + S(y - x, C(y - x))$. $\qquad \square$

**D.2. MIP-encodability of Affine, Conditional, Switch:**

Here we will explain the MIP-encodability each of the affine, conditional, and switch operators. For completeness, we copy the definition of MIP-encodability:

**Definition D.1.** *We say that a function $g$ is **MIP-encodable** if, for every mixed-integer polytope $M$, the image of $M$ mapped through $g$ is itself a mixed-integer polytope.*

We now prove Lemma 2 from the main paper:

**Lemma D.2.** *Let $g$ be a composition of affine, conditional, and switch operators, where global lower and upper bounds are known for each input to each element of the composition. Then $g$ is a MIP-encodable function.*

*Proof.* It suffices to show that each of the primitive operators are MIP-encodable. This amounts to, for each operator $g$, to define a system of linear inequalities $\Gamma(x, x')$ which is satisfied if and only if $g(x) = x'$ (or $x' \in g(x)$ for set valued $g$), provided $x$ lies in the global lower and upper bounds, $x \in [l, u]$.

**Affine Operators:** The affine operator is trivially attainable by letting $\Gamma(x, x')$ be the equality constraint

$$x' = Wx + b \tag{57}$$

**Conditional Operators:** To encode $C(x)$ as a system of linear constraints, we introduce the integer variable $a$ and wish to encode $a = C(x)$, or equivalently, $a = 1 \Leftrightarrow x \geq 0$. We assume that we know values $l, u$ such that $l \leq x \leq u$. Then the implication $a = 1 \Rightarrow x \geq 0$ is encoded by the constraint:

$$x \geq (a - 1) \cdot u \tag{58}$$

Since if $x < 0$, then $a = 1$ yields a contradiction in that $0 > x \geq (1 - 1) \cdot u = 0$. The implication $x \geq 0 \Rightarrow a = 1$ is encoded by the constraint

$$x \leq a \cdot (1 - l) - 1 \tag{59}$$

Since if $x \geq 0$, then $a = 0$ yields a contradiction in that $0 \leq x \leq (0) \cdot (1 - l) - 1 = -1$. Hence $a = 1 \Leftrightarrow x \geq 0$. We note that if $l > 0$ or $u < 0$, then the value of $a$ is fixed and can be encoded with one equality constraint.

**Switch Operators:** Encoding $S(x, a)$ as a system of linear inequalities requires the introduction of continuous variable $y$. As we assume we know $l, u$ such that $l \leq x \leq u$. Denote $\hat{l} := \min(l, 0)$ and $\hat{u} := \max(u, 0)$. The system of linear inequalities $\Gamma(a, x, y)$ is defined as the conjunction of:

$$\begin{array}{ll} y \geq x - u \cdot (1 - a) & y \geq l \cdot a \\ y \leq x - l \cdot (1 - a) & y \leq u \cdot a \end{array} \tag{60}$$

We wish to show that $y = S(x, a) \Leftrightarrow \Gamma(a, x, y)$. Suppose that $\Gamma(a, x, y)$ is satisfied. Then if $a = 1$, $x$ must equal $y$, since it is implied by left-column constraints of equation 60. The right-column constraints are satisfied by assumption. Alternatively, if $a = 0$ then $y$ must equal 0: it is implied by the right-column constraints of equation 60. The left columns are satisfied with $a = 0$ and $y = 1$ since $l \leq x \leq u$ by assumption. On the other hand, suppose $y = S(x, a)$. If $a = 1$, then $y = x$ by definition and we have already shown that $\Gamma(1, x, x)$ satisfied for all $x \in [l, u]$. Similarly, if $a = 0$, then $y = 0$ and we have shown that $\Gamma(0, x, 0)$ is satisfied for all $x \in [l, u]$.

Finally we note that if one can guarantee that $a = 0$ or $a = 1$ always, then only the equality constraint $y = x$ or $y = 0$ is needed. $\square$

**More efficient encodings:** Finally we'll remark that while the above are valid encodings of affine, conditional and switch operators, encodings with fewer constraints for compositions of these primitives do exist. For example, suppose we instead wish to encode a continuous piecewise linear function with one breakpoint over one variable

$$R(x) = \begin{cases} A_1(x) & \text{if } x \geq z \\ A_2(x) & \text{if } x < z \end{cases} \tag{61}$$

for affine funtcions $A_1, A_2 : \mathbb{R} \to \mathbb{R}$, with $A_1(z) = A_2(z)$. Certainly we could use affine, conditional and switch operators, as

$$R(x) = A_1(x) + S\left(A_1(x) - A_2(x), C(z - x)\right) \tag{62}$$

Where which requires 12 linear inequalities. Instead we can encode this function using only 4 linear inequalities. Supposing we know $l, u$ such that $l \leq x \leq u$, then $R(x)$ can be encoded by introducing an auxiliary integer variable $a$ and four constraints. Letting

$$\zeta^- = \min_{x \in [l,u]} A_1(x) - A_2(x) \tag{63}$$

$$\zeta^+ = \max_{x \in [l,u]} A_1(x) - A_2(x) \tag{64}$$

then the constraints $\Gamma(a, x, y)$ are

$$\begin{aligned} y \geq A_1(x) - a\zeta^+ \qquad & y \geq A_2(x) + (1-a)\zeta^- \\ y \leq A_1(x) - a\zeta^- \qquad & y \leq A_2(x) + (1-a)\zeta^+ \end{aligned} \tag{65}$$

This formulation admits a more efficient encoding for functions like $\sigma(\cdot)$, and $|\cdot|$.

### D.3. Abstract Interpretations for Bound Propagation

Here we discuss techniques to compute the lower and upper bounds needed for the MIP encoding of affine, conditional and switch operators. We will only need to show that for each of our primitive operators, we can map sound input bounds to sound output bounds.

For this, we turn to the notion of abstract interpretation. Classically used in static program analysis and control theory, abstract interpretation develops machinery to generate sound approximations for passing sets through functions. This has been used to great success to develop certifiable robustness techniques for neural networks (Singh et al., 2019). Formally, this requires an abstract domain, abstraction and concretization operators, and a pushforward operator for every function we wish to model. An abstract domain $\mathcal{A}^n$ is a family of abstract mathematical objects, each of which *represent* a set over $\mathbb{R}^n$. An abstraction operator $\alpha^n : \mathcal{P}(\mathbb{R}^n) \to \mathcal{A}^n$ maps subsets of $\mathbb{R}^n$ into abstract elements, and a concretization operator $\gamma^n : \mathcal{A}^n \to \mathcal{P}(\mathbb{R})^n$ maps abstract elements back into subsets of $\mathbb{R}^n$. A pushforward operator for function $f : \mathcal{R}^n \to \mathcal{R}^m$, is denoted as $f^\# : \mathcal{A}^n \to \mathcal{A}^m$, and is called *sound*, if for all $\mathcal{X} \subset \mathbb{R}^n$,

$$\{f(x) \mid x \in \mathcal{X}\} \subseteq \gamma^m(f^\#(\alpha^n(\mathcal{X}))) \tag{66}$$

#### D.3.1. HYPERBOXES AND BOOLEAN HYPERBOXES

The simplest abstract domains are the hyperbox and boolean hyperbox domains. The hyperbox abstract domain over $\mathbb{R}^n$ is denoted as $\mathcal{H}^n$. For each $\mathcal{X} \subset \mathbb{R}^n$ such that $H = \alpha(\mathcal{X})$, $H$ is parameterized by two vectors $l, u$ such that

$$l_i \leq \inf_{x \in \mathcal{X}} x_i \qquad u_i \geq \sup_{x \in \mathcal{X}} x_i \tag{67}$$

and $\gamma^n(H) = \{x \mid l \leq x \leq u\}$. An equivalent parameterization is by vectors $c, r$ such that $c = \frac{l+u}{2}$ and $r = \frac{u-l}{2}$. We will sometimes use this parameterization when it is convenient.

Similarly, the boolean hyperbox abstract domain $\mathcal{B}^n$ represents sets over $\{0, 1\}^n$. For each $\mathcal{X}_b \subseteq \{0, 1\}^n$ such that $B = \alpha(\mathcal{X}_b)$, $B$ is parameterized by a vector $v \in \{0, 1, ?\}^n$ such that

$$v_i = \begin{cases} 1 & \text{if } x_i = 1 \quad \forall x \in \mathcal{X}_b \\ 0 & \text{if } x_i = 0 \quad \forall x \in \mathcal{X}_b \\ ? & \text{otherwise} \end{cases} \tag{68}$$

And

$$\gamma^n(B) = \{x \mid (x_i = v_i) \lor (v_i = ?)\} \tag{69}$$

Finally, we can compose these two domains to represent subsets of $\mathbb{R}^n \times \{0, 1\}^m$. For any set $\mathcal{X} \subseteq \mathbb{R}^n \times \{0, 1\}^m$, we let $\mathcal{X}_\mathbb{R}$ refer to the restriction of $\mathcal{X}$ to $\mathbb{R}^n$ and let $\mathcal{X}_b$ refer to the restriction of $\mathcal{X}$ to $\{0, 1\}^m$. Then $\alpha(\mathcal{X}) := (H, B)$ where $H = \alpha(\mathcal{X}_\mathbb{R})$, $B = \alpha(\mathcal{X}_b)$. The concretization operator is defined $\gamma(H, B) := \gamma(H) \times \gamma(B)$.

D.3.2. PUSHFORWARDS FOR AFFINE, CONDITIONAL, SWITCH

We will now define pushforward operators for each of our primitives.

**Affine Pushforward Operators** Provided with bounded set $\mathcal{X} \subset \mathbb{R}^n$, with $H = \alpha(\mathcal{X})$ parameterized by $c, r$, and affine operator $A(x) = Wx + b$, the pushforward $A^\#$ is defined $A^\#(H) = H'$, where $H'$ is parameterized by $c', r'$ with

$$c' = Wc + b \qquad r' = |W|r \tag{70}$$

where $|W|$ is the elementwise absolute value of $W$. To see that this is sound, it suffices to show that for every $x \in \mathcal{X}$, $c'_i - r'_i \leq A(x_i) \leq c'_i + r'_i$. Fix an $x \in \mathcal{X}$ and consider $A(x)_i = w_i^T x + b_i$ where $w_i$ is the $i^{th}$ row of $W$. Note that $x = c + e$ for some vector $e$ with $|e| \leq r$ elementwise. Then

$$
\begin{aligned}
w_i^T x + b_i &= w_i^T c + w_i^T e + b_i \\
&\geq w_i^T c - |w_i^T e| + b_i \\
&\geq w_i^T c - |w_i^T||e| + b_i \\
&\geq w_i^T c - |w_i^T|r + b_i
\end{aligned}
\qquad
\begin{aligned}
w_i^T x + b_i &= w_i^T c + w_i^T e + b_i \\
&\leq w_i^T c + |w_i^T e| + b_i \\
&\leq w_i^T c + |w_i^T||e| + b_i \\
&\leq w_i^T c + |w_i^T|r + b_i
\end{aligned}
$$

as desired.

**Conditional Pushforward Operators** Provided with bounded set $\mathcal{X} \subset \mathbb{R}^n$ with $H = \alpha^n(\mathcal{X})$ parameterized by $l, u$, the elementwise conditional operator is defined $C^\#(H) = B$ where $B$ is parameterized by $v$ with

$$v_i = \begin{cases} 0 & \text{if } u_i < 0 \\ 1 & \text{if } l_i > 0 \\ ? & \text{otherwise} \end{cases} \tag{71}$$

Soundness follows trivially: for any $x \in \mathcal{X}$, if $l_i > 0$, then $x_i > 0$ and $C(x)_i = 1$. If $u_i < 0$, then $x_i < 0$ and $C(x)_i = 0$. Otherwise, $v_i =?$, which is always a sound approximation as $C(x)_i \in \{0, 1\}$.

**Switch Pushforward Operators** Provided with $\mathcal{X} \subset \mathbb{R}^n \times \{0, 1\}^n$, we define $\mathcal{X}_\mathbb{R} := \{x \mid (x, a) \in \mathcal{X}\}$ and $\mathcal{X}_b := \{a \mid (x, a) \in \mathcal{X}\}$. We've defined $\alpha(\mathcal{X}) := (\alpha(\mathcal{X}_\mathbb{R}), \alpha(\mathcal{X}_b))$. Then if $H := \alpha(\mathcal{X}_\mathbb{R})$ parameterized by $l, u$, and $B := \alpha(\mathcal{X}_b)$ parameterized by $v$, we define the pushforward operator for switch $S^\#(H, B) = H'$ where $H'$ is parameterized by $l', u'$ with

$$l'_i = \begin{cases} l_i & \text{if } v_i = 1 \\ 0 & \text{if } v_i = 0 \\ \min(l_i, 0) & \text{otherwise} \end{cases} \tag{72}$$

$$u'_i = \begin{cases} u_i & \text{if } v_i = 1 \\ 0 & \text{if } v_i = 0 \\ \max(u_i, 0) & \text{otherwise} \end{cases} \tag{73}$$

Soundness follows: letting $(x, a) \in \mathcal{X}$, if $v_i = 0$, then $a_i = 0$, and $S(x, a)_i = 0$, hence $l'_i, u'_i = 0$ is sound. If $v_i = 1$, then $a_i = 1$ and $S(x, a)_i = x_i$ and hence $l'_i = l_i, u'_i = u_i$ is sound by the soundness of $H$ over $\mathcal{X}_\mathbb{R}$. Finally, if $v_i =?$, then $a_i = 0$ or $a_i = 1$, and $S(x, a)_i \in \{0\} \cup [l_i, u_i] \subseteq [\min(l_i, 0), \max(u_i, 0)]$.

D.3.3. ABSTRACT INTERPRETATION AND OPTIMIZATION

We make some remarks about the applications of abstract interpretation as a technique for optimization. Recall that, for any set $\mathcal{X}$ and functions $g, f$, if $\mathcal{Y} = \{f(x) \mid x \in \mathcal{X}\}$ we have that

$$\max_{x \in \mathcal{X}} g(f(x)) = \max_{y \in \mathcal{Y}} g(y) \tag{74}$$

Instead if $\mathcal{Z}$ is such that $\{f(x) \mid x \in \mathcal{X}\} \subseteq \mathcal{Z}$, then

$$\max_{x \in \mathcal{X}} g(f(x)) \leq \max_{z \in \mathcal{Z}} g(z) \tag{75}$$

In particular, suppose $f$ is a nasty function, but $g$ has properties that make it amenable to optimization. Optimization frameworks may not be able to solve $\max_{x \in \mathcal{X}} g(f(x))$. On the other hand, it might be the case that the RHS of equation 75 is solvable. In particular, if $g$ is concave and $\mathcal{Z}$ is a convex set obtained by $\mathcal{Z} := \gamma(f^{\#}(\alpha(\mathcal{X})))$, then by soundness we have $\mathcal{Z} \supset \mathcal{Y}$. In fact, this is the formal definition of a convex relaxation.

Under this lens, one can use the abstract domains and pushforward operators previously defined to recover FastLip (Weng et al., 2018a), though the algorithm was not presented using abstract interpretations. Indeed, using the hyperbox and boolean hyperbox domains, over a set $\mathcal{X}$, one can recover a hyperbox $\mathcal{Z} \supseteq \{\nabla f(x) \mid x \in \mathcal{X}\}$. Then we have that

$$\max_{x \in \mathcal{X}} ||\nabla f(x)|| \leq \max_{z \in \mathcal{Z}} ||z|| \tag{76}$$

where it is easy to optimize $\ell_p$-norms over hyperboxes. In addition, many convex-relaxation approaches towards certifiable robustness may be recovered by this framework (Singh et al., 2019; Zico Kolter and Wong, 2017; Raghunathan et al., 2018; Zhang et al., 2018).

# E. Extensions of LipMIP

This section will provide more details regarding how we extend LipMIP to be applicable to vector-valued functions and to other norms. We will present an example of a nonstandard norm by detailing an application towards untargeted classification robustness.

### E.1. Extension of LipMIP to Vector-Valued Networks

Letting $f : \mathbb{R}^n \to \mathbb{R}^m$ be a vector-valued ReLU network in general position, suppose $|| \cdot ||_\alpha$ is a norm over $\mathbb{R}^n$ and $|| \cdot ||_\beta$ is a norm over $\mathbb{R}^m$. Further, suppose $\mathcal{X}$ is some open subset of $\mathbb{R}^n$. Then Theorem 1 states that

$$L^{(\alpha,\beta)}(f, \mathcal{X}) := \sup_{x \neq y \in \mathcal{X}} \frac{||f(x) - f(y)||_\beta}{||x - y||_\alpha} = \sup_{G \in \delta_f(\mathcal{X})} ||G^T||_{\alpha,\beta}. \tag{77}$$

And noting that

$$||G^T||_{\alpha,\beta} := \sup_{||y||_\alpha \leq 1} \sup_{||z||_{\beta*} \leq 1} |zG^T y| \tag{78}$$

one can substitute Equation 78 into Equation 77 to yield

$$L^{(\alpha,\beta)}(f, \mathcal{X}) = \sup_{G \in \delta_f(\mathcal{X})} \sup_{||y||_\alpha \leq 1} \sup_{||z||_{\beta*} \leq 1} |zG^T y| \tag{79}$$

The key idea is that we can define function $g_z : \mathbb{R}^n \to \mathbb{R}$ as

$$g_z(x) = \langle z, f(x) \rangle \tag{80}$$

Where

$$\delta_{g_z}(x)^T = \{Gz \mid G \in \delta_f(x)\} \tag{81}$$

The plan is to make LipMIP optimize over $x$ and $z$ simultaneously and maximize the gradient norm of $g_z(x)$. To be more explicit, we note that the scalar-valued LipMIP solves::

$$\sup_{x \in \mathcal{X}} ||\nabla^\# f(x)^T||_{\alpha,|\cdot|} = \sup_{G \in \nabla^\# f \mathcal{X}} ||G||_{\alpha*} = \sup_{x \in \nabla^\# f(\mathcal{X})} \sup_{||y||_\alpha \leq 1} |\nabla^\# f(x)^T y| \tag{82}$$

where we have shown that $\nabla f(x)$ is MIP-encodable and the supremum over $y$ can be encoded for $|| \cdot ||_1, || \cdot ||_\infty$, because there exist nice closed form representations of $|| \cdot ||_1, || \cdot ||_\infty$. The extension, then, only comes from the $\sup_{||z||_{\beta*} \leq 1}$ term. We can explicitly define $f$ as

$$f(x) = W_{d+1}\sigma(Z_d(x)) \qquad Z_i(x) = W_i\sigma(Z_{i-1}(x)) + b_i \qquad Z_0(x) = x \tag{83}$$

such that

$$g_z(x) = z^T W_{d+1}\sigma(Z_d(x)) \qquad Z_i(x) = W_i\sigma(Z_{i-1}(x)) + b_i \qquad Z_0(x) = x \tag{84}$$

And the recursion for $\nabla^\# g_z(x)$ is defined as

$$\nabla^\# g_z(x) = W_1^T Y_1(x) \qquad Y_i(x) = W_{i+1}^T \text{Diag}(\Lambda_i(x))Y_{i+1}(x) \qquad Y_{d+1}(x) = W_{d+1}^T z \tag{85}$$

Thus we notice the only change occurs in the definition of $Y_{d+1}(x)$. In the scalar-valued $f$ case, $Y_{d+1}(x)$ is always the constant vector, $c$. In the vector-valued case, we can let $Y_{d+1}(x)$ be the output of an affine operator. Thus as long as the dual ball $\{z \mid ||z||_\beta^*\}$ is representable as a mixed-integer polytope, we may solve the optimization problem of Equation 79.

**Definition E.1.** *We say that a norm $|| \cdot ||_\beta$ is a linear norm over $\mathbb{R}^k$ if the set of points such that $||z||_{\beta*} \leq 1$ is representable as a mixed-integer polytope.*

Certainly the $\ell_1, \ell_\infty$ norms are linear norms, but we will demonstrate another linear norm in the next subsection.

**Corollary E.1.** *In the same setting as Theorem 5, if $|| \cdot ||_\alpha$ is $|| \cdot ||_1$ or $|| \cdot ||_\infty$, and $|| \cdot ||_\beta$ is a linear norm, then LipMIP applied to $f$ and $\mathcal{X}$ yields the answer*

$$L^{(\alpha, \beta)}(f, \mathcal{X}) \tag{86}$$

*where the parameters of LipMIP have been adjusted to reflect the norms of interest.*

*Proof.* The proof ideas are identical to that for Theorem 5. The only difference is that the norm $|| \cdot ||_\beta$ has been replaced from $| \cdot |$ to an arbitrary linear norm. The argument for correctness in this case is presented in the paragraphs preceding the corollary statement. $\square$

### E.2. Application to Untargeted Classification Robustness

Now we turn our attention towards untargeted classification robustness. In the binary classification setting, we let $f : \mathbb{R}^n \to \mathbb{R}$ be a scalar-valued ReLU network. Then the label that classifier $f$ assigns to point $x$ is $\text{sign}(f(x))$. In this case, it is known that for any open set $\mathcal{X}$, any norm $|| \cdot ||_\alpha$, any $x, y \in \mathcal{X}$,

$$||x - y||_\alpha < \frac{|f(x)|}{L^\alpha(f, \mathcal{X})} \implies \text{sign}(f(y)) = \text{sign}(f(x)) \tag{87}$$

Indeed, this follows from the definition of the Lipschitz constant as

$$L^\alpha(f, \mathcal{X}) := \sup_{x \neq y} \frac{|f(x) - f(y)|}{||x - y||_\alpha}. \tag{88}$$

Then, by the contrapositive of implication 87 , if $\text{sign}(f(x)) \neq \text{sign}(f(y))$ then $|f(x) - f(y)| \geq |f(x)|$ and for all $x, y \in \mathcal{X}$,

$$L^\alpha(f, \mathcal{X}) \geq \frac{|f(x) - f(y)|}{||x - y||_\alpha} \tag{89}$$

Rearranging, we have

$$||x - y||_\alpha \geq \frac{|f(x) - f(y)|}{L^\alpha(f, \mathcal{X})} \geq \frac{|f(x)|}{L^\alpha(f, \mathcal{X})} \tag{90}$$

arriving at the desired contrapositive implication.

In the multiclass classification setting, we introduce the similar lemma.

**Lemma E.1.** $f : \mathbb{R}^n \to \mathbb{R}^m$ *assigns the label as the index of the maximum logit. We will define the hard classifier* $F : \mathbb{R}^n \to [m]$ *as* $F(x) = \arg\max_i f(x)_i$. *We claim that for any* $\mathcal{X}$, *and norm* $|| \cdot ||_\alpha$, *if* $F(x) = i$, *then for all* $y \in \mathcal{X}$,

$$||x - y||_\alpha < \min_j \frac{|f_{ij}(x)|}{L^\alpha(f_{ij}, \mathcal{X})} \implies F(y) = i \tag{91}$$

*where we've defined* $f_{ij}(x) := (e_i - e_j)^T f(x)$.

*Proof.* To see this, suppose $F(y) = j$ for some $j \neq i$. Then $|f_{ij}(x) - f_{ij}(y)| \geq |f_{ij}(x)|$, as by definition $f_{ij}(x) > 0$ and $f_{ij}(y) < 0$. Then by the definition of Lipschitz constant :

$$L^\alpha(f_{ij}, \mathcal{X}) \geq \frac{|f_{ij}(x) - f_{ij}(y)|}{||x - y||_\alpha} \geq \frac{|f_{ij}(x)|}{||x - y||_\alpha} \tag{92}$$

arriving at the desired contrapositive LHS. We only note that we need to take $\min$ over all $j$ so that $f_{ij}(y) \geq 0$ for all $j$. $\square$

Now we present our main Theorem regarding multiclassification robustness:

**Theorem E.1.** *Let $f$ be a vector-valued ReLU network, and let $|| \cdot ||_\times$ be a norm over $\mathbb{R}^m$, such that for any $x$ in any open set $\mathcal{X}$, with $F(x) = i$,*

$$\min_j \frac{|f_{ij}(x)|}{L^{(\alpha,\times)}(f,\mathcal{X})} \leq \min_j \frac{|f_{ij}(x)|}{L^\alpha(f_{ij},\mathcal{X})}. \tag{93}$$

*Then for $x, y \in \mathcal{X}$ with $F(x) = i$,*

$$||x - y||_\alpha < \min_j \frac{|f_{ij}(x)|}{L^{(\alpha,\times)}(f,\mathcal{X})} \implies F(y) = i. \tag{94}$$

*In addition, if $|| \cdot ||_\times$ is a linear norm, and $f$ is in general position, then $L^{(\alpha,\times)}(f,\mathcal{X})$ is computable by LipMIP.*

*Proof.* Certainly equation 94 follows directly from Lemma E.1 and equation 93. $\qquad\square$

What remains to be shown is a $|| \cdot ||_\times$ such that equation 93 holds. To this end, we present a lemma describing convenient formulations for norms:

**Lemma E.2.** *Let $\mathcal{C} \subseteq \mathbb{R}^n$ be a set that contains an open set. Then*

$$||x||_\mathcal{C} := \sup_{y \in \mathcal{C}} |y^T x| \tag{95}$$

*is a norm.*

*Proof.* Nonnegativity and absolute homogeneity are trivial. To see the triangle inequality holds for $|| \cdot ||_\mathcal{C}$, we see that, for any $x, y$,

$$||x + y||_\mathcal{C} := \sup_{z \in \mathcal{C}} |z^T(x + y)| \leq \sup_{z \in \mathcal{C}} |z^T x| + \sup_{z' \in \mathcal{C}} |z'^T y| \leq ||x||_\mathcal{C} + ||y||_\mathcal{C} \tag{96}$$

And point separation follows because $\mathcal{C}$ contains an open set and if $x \neq 0$, then there exists at least one $y$ in $\mathcal{C}$ such that $|y^T x| > 0$. $\qquad\square$

Now we can define our norm $|| \cdot ||_\times$ that satisfies equation 93:

**Definition E.2.** *Let $e_{ij} := e_i - e_j$ where each $e_i$ is the elementary basis vector in $\mathbb{R}^m$. Then let $\mathcal{E}$ be the convex hull of all such $e_{ij}$ and all $e_i$, $\mathcal{E} := Conv(\{e_i \mid i \in [m]\} \cup \{e_{ij} \mid i \neq j \in [m]\})$. We define the **cross-norm**, $|| \cdot ||_\times$ as*

$$||x||_\times := \sup_{y \in \mathcal{E}} |y^T x| \tag{97}$$

We note that by Lemma E.2 and since $\mathcal{E}$ contains the positive simplex, $\mathcal{E}$ contains an open set and hence the cross-norm is certainly a norm. Indeed, because the convex hull of a finite point-set is a polytope, the cross-norm is a linear norm. Further, we note that the polytope $\mathcal{E}$ has an efficient H-description.

**Proposition E.1.** *The set $\mathcal{E} \subseteq \mathbb{R}^m$, is equivalent to the polytope $\mathcal{P}$ defined as*

$$\mathcal{P} = \left\{ x \;\middle|\; \begin{array}{l} x = x^+ - x^- \\ x^+ \geq 0 \quad ; \quad \sum_i x_i^+ \leq 1 \\ x^- \geq 0 \quad ; \quad \sum_i x_i^- \leq 1 \\ \sum_i x_i^+ \geq \sum_i x_i^- \end{array} \right\} \tag{98}$$

*Proof.* As $\mathcal{E}$ is the convex hull of $e_{ij}$ and $e_i$ for all $i \neq j \in [m]$. Certainly each of these points is feasible in $\mathcal{P}$, and since $\mathcal{P}$ is convex, by the definition of a convex hull, $\mathcal{E} \subseteq \mathcal{P}$. In the other direction, consider some $x \in \mathcal{P}$. Decompose $x$ into $x^+$, and $-x^-$, by only considering the positive and negative components of $x$. The goal is to write $x$ as a convex combination of $\{e_{ij}, e_i\}$. Further decompose $x^+$ into $y^+, z^+$ such that $x^+ = y^+ + z^+$, $y^+ \geq 0$, $z^+ \geq 0$, and $\sum_i y_i^+ = \sum_i x_i^-$. Then we can write $y_i^+ - x_i^-$ as a convex combination of $e'_{ij}s$ and $z_i^+$ is a convex combination of $e'_i s$ and 0, where we note that $0 \in \mathcal{E}$ because $e_{ij}, e_{ji}$ are in $\mathcal{E}$. $\qquad\square$

Now we desire to show equation 93 holds for the cross-norm.

**Proposition E.2.** *For any open set $\mathcal{X}$, the Lipschitz constant with respect to the cross norm, $||\cdot||_\times$, and any norm $||\cdot||_\alpha$, for $x \in \mathcal{X}$ with $F(x) = i$, then*

$$\min_j \frac{|f_{ij}(x)|}{L^{(\alpha,\times)}(f,\mathcal{X})} \leq \min_j \frac{|f_{ij}(x)|}{L^\alpha(f_{ij},\mathcal{X})}. \tag{99}$$

To see this, notice

$$\frac{\min_j |f_{ij}(x)|}{\max_j L^\alpha(f_{ij},\mathcal{X})} \leq \min_j \frac{|f_{ij}(x)|}{L^\alpha(f_{ij},\mathcal{X})}. \tag{100}$$

so it amounts to show that $L^{(\alpha,\times)}(f,\mathcal{X}) \geq \max_j L^\alpha(f_{ij},\mathcal{X})$. By the definition of the Lipschitz constant:

$$\max_j L^\alpha(f_{ij},\mathcal{X}) = \max_j \sup_{x \neq y \in \mathcal{X}} \frac{|\langle e_i - e_j, f(x) - f(y)\rangle|}{||x - y||_\alpha} \tag{101}$$

By switching the sup and max above, and observing that, for all $z$,

$$\max_j |\langle e_i - e_j, z\rangle| \leq ||z||_\times \tag{102}$$

we can bound

$$\max_j L^\alpha(f_{ij},\mathcal{X}) \leq \frac{||f(x) - f(y)||_\times}{||x - y||_\alpha} \leq L^{(\alpha,\times)}(f,\mathcal{X}) \tag{103}$$

as desired.

# F. Experiments

In this section we describe details about the experimental section of the main paper and present additional experimental results.

## F.1. Experimental Setup

**Computing environment:**   All experiments were run on a desktop with an Intel Core i7-9700K 3.6 GHz 8-Core Processor and 64GB of RAM. All experiments involving mixed-integer or linear programming were optimized using Gurobi, using two threads maximum (Gurobi Optimization, 2020).

**Synthetic Datasets:**   The main synthetic dataset used in our experiments is generated procedurally with the following parameters:

$$\{\texttt{dim, num\_points, min\_separation, num\_classes, num\_leaders}\}$$

The procedure is as follows: randomly sample `num_points` from the unit hypercube in `dim` dimensions. Points are sampled sequentially, and a sample is replaced if it is within `min_separation` of another, previously sampled point. Next, `num_leaders` points are selected uniformly randomly from the set of points and uniformly randomly assigned a label from 1 to `num_classes`. The remaining points are labeled according to the label of their closest 'leader'. An example dataset and classifier learned to classify it are presented in Figure 3.

*Figure 3.* (Left): Synthetic dataset used for evaluating effect of training on Lipschitz Estimation. (Right): Decision boundaries of a neural network trained using only CrossEntropy loss for 1000 epochs on the synthetic dataset.

**Estimation Techniques:**   Here we will outline the hyperparameters and computing environment for each estimation technique compared against.

- **RandomLB:** We randomly sample 1000 points in the domain of interest. At each point, we evaluate the appropriate gradient norm that lower-bounds the Lipschitz constant. We report the maximum amongst these sampled gradient norms.

- **CLEVER:** We randomly sample 500 batches of size 1024 each and compute the appropriate gradient norm for each, for a total of 512,000 random gradient norm evaluations. The hyperparameters used to estimate the best-fitting Reverse Weibull distribution are left to their defaults from the CLEVER Github Repository: https://github.com/IBM/CLEVER-Robustness-Score, (Weng et al., 2018b).

- **LipMIP:** LipMIP is evaluated exactly without any early stopping or timeout parameters, using 2 threads and the Gurobi optimizer.

- **LipSDP:** We use the LipSDP-Network formulation outlined in Fazlyab et al. (2019), which is the slowest but tightest formulation. We note that since this only provides an upper bound of the $\ell_2$-norm of the gradient. We scale by a factor of $\sqrt{n}$ for $\ell_1, \ell_\infty$ estimates over domains that are subsets of $\mathbb{R}^n$.

- **SeqLip:** SeqLip bounds are attained by splitting each network into subproblems, with one subproblem per layer. The $||\cdot||_{2,2}$ norm of the Jacobian of each layer is estimated using the *Greedy SeqLip* heuristic (Virmaux and Scaman, 2018). We scale the resulting output by a factor of $\sqrt{d}$. We remark that a cheap way to make this technique local would be to use interval analysis over a local domain to determine which neurons are fixed to be on or off, and do not include decision variables for these neurons in the optimization step.

- **FastLip:** We use a custom implementation of FastLip that more deeply represents the abstract interpretation view of this technique. As we have noted several times throughout this paper, this is equivalent to the FastLip formulation of Weng et al. (2018a).

- **NaiveUB:** This naive upper bound is generated by multiplying the operator norm of each affine layer's Jacobian matrix and scaling by $\sqrt{d}$.

## F.2. Experimental Details

Here we present more details about each experiment presented in the main paper.

**Accuracy vs. Efficiency Experiments**   In the random dataset example, we evaluated 20 randomly generated neural networks with layer sizes $[16, 16, 16, 2]$. Parameters were initialized according to He initialization (He et al., 2015).

In the synthetic dataset example, a random dataset with 2000 points over $\mathbb{R}^{10}$, 20 leaders 2 classes, were used to train 20 networks, each of layer sizes $[10, 20, 30, 30, 2]$ and was trained for 500 epochs using CrossEntropy loss with the Adam optimizer with learning rate 0.001 and no weight decay (Kingma and Ba, 2014).

In the MNIST example, only MNIST 1's and 7's were selected for our dataset. We trained 20 random networks of size $[784, 20, 20, 20, 2]$. We trained for 10 epochs using the Adam optimizer, with a learning rate of 0.001, where the loss was the CrossEntropy loss and an $\ell_1$ weight decay regularization term with value $1 \cdot 10^{-5}$. For each of these networks, 20 randomly centered $\ell_\infty$ balls of radius 0.1 were evaluated.

For each experiment, we presented only the results for compute time and standard deviations, as well as mean relative error with respect to the answer returned by LipMIP.

**Effect of Training on Lipschitz Constant**   When demonstrating the effect of different regularization schemes we train a 2-dimensional network with layer sizes $[2, 20, 40, 20, 2]$ over a synthetic dataset generated using 256 points over $\mathbb{R}^2$, 2 classes and 20 leaders. All training losses were optimized for 1000 epochs using the Adam optimizer with a learning rate of 0.001, and no implicit weight decay. Snapshots were taken every 25 epochs and LipMIP was evaluated over the $[0, 1]^2$ domain. All loss functions incorporated the CrossEntropy loss with a scalar value of 1.0. The FGSM training scheme replaced all clean examples with adversarial examples generated via FGSM and a step size of 0.1. The $\ell_1$-weight regularization scheme had a penalization weight of $1 \cdot 10^{-4}$ and the $\ell_2$-weight regularization scheme had a penalization weight of $1 \cdot 10^{-3}$. Weights for $\ell_p$ weight penalties were chosen to not affect training accuracy and were determined by a line search.

The same setup was used to evaluate the accuracy of various estimators during training. The network that was considered in this case was the one trained only using CrossEntropy loss.

**Random Networks and Lipschitz Constants**   For the random network experiment, 5000 neural networks with size $[10, 10, 10, 1]$ were initialized using He initialization. LipMIP evaluated the maximal $\ell_1$ norm of the gradient over an origin-centered $\ell_\infty$ ball of radius 1000.

*Figure 4.* (Left): Evaluation of Lipschitz estimators over changing width. A fixed dataset and training scheme is picked, and multiple networks with varying width are trained. Notice the small increase in Lipschitz constant returned by LipMIP, while the ever-increasing absolute error between efficient estimation techniques. (Right): Evaluation of Lipschitz estimators over changing depth. A fixed dataset and training scheme is picked, and multiple networks with varying depth are trained. Note that the $y$-axis is a *log-scale*, implying that Lipschitz constants are much more sensitive to depth than width

### F.3. Additional Experiments

**Effect of Architecture on Lipschitz Estimation**   We investigate the effects of changing architecture on Lipschitz estimation techniques. We generate a single synthetic dataset, train networks with varying depth and width, and evaluate each Lipschitz Estimation technique on each network over the $[0, 1]^2$ domain. The synthetic dataset used is over 2 dimensions, with 300 random points, 10 leaders and 2 classes. Training for both the width and depth series is performed using 200 epochs of Adam with learning rate 0.001 over the CrossEntropy loss, with no regularization.

To investigate the effects of changing width, we train networks with size $[2, C, C, C, 2]$ where $C$ is the x-axis displayed in Figure 4 (left).

To explore the effects under changing depth, we train networks with size $[2] + [20] \times C + [2]$, where $C$ is the x-axis displayed in Figure 4 (right). Note that the $y$-axis is a *log-scale*: indicating that estimated Lipschitz constants rise exponentially with depth.

**Estimation of $L^1(f, \mathcal{X})$:**   Experiments presented in the main paper evaluate $L^\infty(f, \mathcal{X})$, which is the maximal $\ell_1$ norm of the gradient. We can present the same experiments over the $\ell_\infty$ norm of the gradient. Tables 1 and 2 display results comparing estimation techniques for $L^1(f, \mathcal{X})$ under the same settings as the "Accuracy vs. Efficiency" experiments in the main paper: that is, we estimate the maximal $||\nabla f||_\infty$ over random networks, networks trained on synthetic datasets, and MNIST networks. Figure 5 demonstrates the effects of training and regularization on Lipschitz estimators on $L^1(f, \mathcal{X})$. Figure 6 plots a histogram of both $L^1(f)$ and $L^\infty(f)$ over random networks.

**Relaxed LipMIP**   In section 7 of the main paper, we described two relaxed forms of LipMIP: one that leverages early stopping of mixed-integer programs that can be terminated at a desired integrality gap, and one that is a linear-programming relaxation of LipMIP. Here we present results regarding the accuracy vs. efficiency tradeoff for these techniques. We evaluate LipMIP with integrality gaps of at most $\{100\%, 10\%, 1\%, 0\%\}$ and LipLP over the unit hypercube on the same random networks and synthetic datasets used to generate the data in Table 1. These results are displayed in Table 3.

**Cross-Lipschitz Evaluation**   We evaluate the $|| \cdot ||_\times$ norm for applications in untargeted robustness verification. We generate a synthetic dataset over 8 dimensions, with 2000 data points, 200 leaders, and 100 classes. We run 10 trials of the following procedure: train a network with layer sizes $[8, 40, 40, 40, 100]$ and pick 20 random data points to evaluate over an $\ell_\infty$ ball of radius 0.1. We train the network with CrossEntropy loss, $\ell_1$-regularization with constant $5 \cdot 10^{-4}$ and train for 2000 epochs using Adam with a learning rate of 0.001 and no other weight-decay terms. Accuracy is at least 65% for each trained network. We evaluate the time and reported Lipschitz value for the following metrics, for data points that have label

| | Random Network | | Synthetic Dataset | |
|---|---|---|---|---|
| Method | Time (s) | Relative Error | Time (s) | Relative Error |
| **RandomLB** | $0.238 \pm 0.004$ | $-30.43\%$ | $0.301 \pm 0.004$ | $-29.27\%$ |
| **CLEVER** | $1.442 \pm 0.071$ | $-13.00\%$ | $55.847 \pm 79.212$ | $0.00\%$ |
| **LipMIP** | $18.825 \pm 19.244$ | $0.00\%$ | $1.873 \pm 0.030$ | $4.18\%$ |
| **FastLip** | $0.001 \pm 0.000$ | $167.55\%$ | $0.028 \pm 0.001$ | $156.67\%$ |
| **LipLP** | $0.018 \pm 0.009$ | $167.55\%$ | $0.001 \pm 0.000$ | $156.67\%$ |
| **LipSDP** | $2.624 \pm 0.026$ | $559.97\%$ | $2.705 \pm 0.030$ | $432.47\%$ |
| **SeqLip** | $0.007 \pm 0.001$ | $773.38\%$ | $0.015 \pm 0.002$ | $674.84\%$ |
| **NaiveUB** | $0.000 \pm 0.000$ | $3121.51\%$ | $0.000 \pm 0.000$ | $11979.62\%$ |

*Table 1.* Accuracy vs. Efficiency tradeoffs for estimating $L^1(f, \mathcal{X})$ over the unit hypercube on random networks of layer sizes $[16, 16, 16, 2]$ and networks with layer sizes $[10, 20, 30, 2]$ trained over a synthetic dataset.

| | Radius 0.1 | | Radius 0.2 | |
|---|---|---|---|---|
| Method | Time (s) | Relative Error | Time (s) | Relative Error |
| **RandomLB** | $0.330 \pm 0.006$ | $-44.25\%$ | $0.325 \pm 0.005$ | $-35.08\%$ |
| **CLEVER** | $6.855 \pm 4.984$ | $-38.31\%$ | $23.010 \pm 0.612$ | $-27.24\%$ |
| **LipMIP** | $4.550 \pm 2.519$ | $0.00\%$ | $4.292 \pm 1.183$ | $0.00\%$ |
| **FastLip** | $0.001 \pm 0.000$ | $+43.45\%$ | $0.233 \pm 0.012$ | $+32.66\%$ |
| **LipLP** | $0.229 \pm 0.021$ | $+43.45\%$ | $0.002 \pm 0.001$ | $+32.66\%$ |
| **NaiveUB** | $0.000 \pm 0.000$ | $+961.31\%$ | $0.000 \pm 0.000$ | $+891.10\%$ |
| **LipSDP** | $18.184 \pm 1.935$ | $+16147.15\%$ | $20.161 \pm 2.333$ | $+14526.92\%$ |
| **SeqLip** | $0.013 \pm 0.004$ | $+16559.65\%$ | $0.021 \pm 0.004$ | $+14917.94\%$ |

*Table 2.* Accuracy vs. Efficiency for estimating local $L^1(f)$ Lipschitz constants on a network with layer sizes $[784, 20, 20, 20, 2]$ trained to distinguish between MNIST 1's and 7's. We evaluate the local lipschitz constant where $\mathcal{X}$'s are chosen to be $\ell_1$-balls with specified radius centered at random points in the unit hypercube.

*Figure 5.* (Left) Effect of training on various Lipschitz estimators in the $L^1(f, \mathcal{X})$ setting. A network of layer sizes $[2, 20, 40, 20, 2]$ was trained using Adam to minimize CrossEntropy loss over a synthetic dataset. Notice how in this setting, even LipSDP does not provide a tight bound. (Right) Effect of regularization scheme on $L^1(f, \mathcal{X})$.

*Figure 6.* Histograms for $L^1(f)$ and $L^\infty(f)$ over random networks with layer sizes $[10, 10, 10, 1]$. Notice the much tighter concentration for $L^1(f)$.

| | Random Network | | | Synthetic Dataset | | |
|---|---|---|---|---|---|---|
| **Method** | **Time (s)** | **Value** | **Relative Error** | **Time (s)** | **Value** | **Relative Error** |
| **LipLP** | $0.017 \pm 0.003$ | 5.508 | $+462.26\%$ | $0.032 \pm 0.003$ | 2087.957 | $+389.80\%$ |
| **LipMIP(100%)** | $132.988 \pm 119.857$ | 1.937 | $+91.98\%$ | $14.690 \pm 12.773$ | 742.395 | $+69.91\%$ |
| **LipMIP(10%)** | $355.974 \pm 294.666$ | 1.102 | $+9.37\%$ | $50.931 \pm 53.403$ | 484.550 | $+8.75\%$ |
| **LipMIP(1%)** | $357.620 \pm 287.511$ | 1.015 | $+0.72\%$ | $59.969 \pm 63.484$ | 448.872 | $+0.57\%$ |
| **LipMIP** | $362.533 \pm 304.685$ | 1.009 | $0.00\%$ | $60.123 \pm 63.721$ | 446.327 | $0.00\%$ |

*Table 3.* Performance vs. accuracy evaluations of various relaxations of LipMIP. LipLP is the linear programming relaxation, and LipMIP(x%) refers to early stopping of LipMIP once an integrality gap of x% has been attained. It can be significantly more efficient to attain reasonable upper bounds than it is to compute the Lipschitz constant exactly.

| Method | Time (s) | Value | Relative Error |
|---:|:---:|:---:|---:|
| **Naive** | $97.698 \pm 104.281$ | 72.083 | 0.00% |
| **CrossLip(i)** | $6.156 \pm 14.256$ | 350.176 | +441.14% |
| **MIPCrossLip(i)** | $6.987 \pm 14.704$ | 350.176 | +441.14% |

*Table 4.* Application to multiclass robustness verification. On networks trained on a synthetic dataset with 100 classes, we evaluate untargeted robustness verification techniques across random datasets. The value column refers to the computed Lipschitz value, and relative error is relative to the Naive value. Notice that a 15x speedup is attainable on average at the cost of providing a 4.5x looser bound on robustness.

$i$:

$$\min_j L^{\infty}(f_{ij}, \mathcal{X}), \tag{104}$$

where we evaluate this naively (**Naive**), or with the search space of all $e_{ij}$ encoded directly with mixed-integer-programming (**MIPCrossLip(i)**). We also evaluate the $|| \cdot ||_{\times(i)}$ norm in lieu of $|| \cdot ||_{\beta}$, where $|| \cdot ||_{\times(i)}$ is defined as

$$||x||_{\times(i)} := \sup_{y \in \mathcal{P}(i)} |y^T x| \tag{105}$$

$$\mathcal{P}(i) := \text{Conv}\left(\{e_i - e_j \mid j \in [m]\} \cup \{e_i \mid i \in [m]\}\right) \tag{106}$$

and we evaluate

$$L^{(\infty, ||\cdot||_{\times(i)})}(f, \mathcal{X}) \tag{107}$$

where we denote this technique (**CrossLip(i)**). Times and returned Lipschitz values are displayed in Table 4.

## Footnotes

[1] We typically write the Jacobian of a function $f : \mathbb{R}^d \to \mathbb{R}^m$ as $\nabla f(x)^T \in \mathbb{R}^{m \times n}$. This is because we like to think of the Jacobian of a scalar-valued function, referred to as the gradient and denoted as $\nabla f(x)$, as a vector/column-vector

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

[Supplementary Material 2 · poster.pdf]



# BACKGROUND & SETTING

## 1. Abstract

In this paper, we estimate and analyze the **Lipschitz constant of neural networks**, key metric for safe and robust deep learning. First, we show that even for two layer neural networks, **exact computation is NP-hard**. Then, we provide **two Lipschitz upper bounds**: *AutoLip*, the first generic algorithm for upper bounding the Lipschitz constant of any automatically differentiable function, and *SeqLip*, an improved algorithm for MLPs. Moreover, we provide an **AutoGrad compliant power method** algorithm, allowing efficient computations even on large convolutions. Our experiments show that *SeqLip* can significantly improve on the existing upper bounds.

## 2. Notations

1. A function $f : \mathbb{R}^n \to \mathbb{R}^m$ is *Lipschitz continuous* if
$$\forall x, y \in \mathbb{R}^n, \ \|f(x) - f(y)\|_2 \leq L \|x - y\|_2.$$
The smallest such $L$ is the *Lipschitz constant* of $f$, denoted $L(f)$.

2. A $K$-layer *Multi-Layer-Perceptron* $f_{MLP} : \mathbb{R}^n \to \mathbb{R}^m$ is the function
$$f_{MLP}(x) = T_K \circ \rho_{K-1} \circ \cdots \circ \rho_1 \circ T_1(x),$$
where $T_k(x) = M_k x + b_k$ is affine and $\rho_k(x) = (g_k(x_i))_{i \in [\![1, n_k]\!]}$ non-linear.

## 3. Exact Lipschitz computation is NP-hard

Problem **LIP-CST**:

**Input:** Two matrices $M_1 \in \mathbb{R}^{l \times n}$ and $M_2 \in \mathbb{R}^{m \times l}$, and a constant $\ell \geq 0$.

**Question:** Let $f = M_2 \circ \rho \circ M_1$ where $\rho(x) = \max\{0, x\}$ is the ReLU activation function.
Is the Lipschitz constant $L(f) \leq \ell$ ?

**Theorem LIP-CST** is NP-hard.
Reduction to maximizing a convex quadratic function on the hypercube.

## 4. Lipschitz constant of affine layers

**Algorithm 1** AutoGrad compliant power method

**Input:** affine function $f : \mathbb{R}^n \to \mathbb{R}^m$, number of iteration $N$

**Output:** approximation of the Lipschitz constant $L(f)$

1: **for** $k = 1$ to $N$ **do**
2:    $v \leftarrow \nabla g(v)$ where $g(x) = \frac{1}{2}\|f(x) - f(0)\|_2^2$
3:    $\lambda \leftarrow \|v\|_2$
4:    $v \leftarrow v/\lambda$
5: **end for**
6: **return** $L(f) = \|f(v) - f(0)\|_2$

**Other layers** Most common other layers in neural networks including activation functions, pooling, batch normalization have simple and explicit Lipschitz constants.

# UPPER BOUNDS

## 5. AutoLip

For sequential networks, this upper bounds is simply the product of spectral norms. Otherwise, we propagate the Lipschitz upper bound with Alg. 2.

Figure: Example of a computation graph for $f_\omega(x) = \ln(1 + e^{x/2}) + |x/2 - \omega \sin(x)|$

**Algorithm 2 AutoLip**

**Input:** function $f : \mathbb{R}^n \to \mathbb{R}^m$ and its computation graph $(g_1, ..., g_K)$

**Output:** upper bound on the Lipschitz constant: $\hat{L}_{AL} \geq L(f)$

1: $\mathcal{Z} = \{(z_0, ..., z_K) : \forall k \in [\![0, K]\!], \theta_k \text{ is constant} \Rightarrow z_k = \theta_k(0)\}$
2: $L_0 \leftarrow 1$
3: **for** $k = 1$ to $K$ **do**
4:    $L_k \leftarrow \sum_{i=1}^{k-1} \max_{z \in \mathcal{Z}} \|\partial_i g_k(z)\|_2 L_i$
5: **end for**
6: **return** $\hat{L}_{AL} = L_k$

## 6. SeqLip

**Idea:** Exploit the relationships between singular vectors of consecutive layers.

$$L(f_{MLP}) \leq \max_{\forall i, \ \sigma_i \in [0,1]^{n_i}} \|M_K \operatorname{diag}(\sigma_{K-1}) \cdots \operatorname{diag}(\sigma_1) M_1\|_2,$$
$$\leq \max_{\forall i, \ \sigma_i \in [0,1]^{n_i}} \|\Sigma_K V_K^\top \operatorname{diag}(\sigma_K) U_{K-1} \Sigma_{K-1} V_{K-1}^\top \operatorname{diag}(\sigma_{K-1}) \cdots \|_2$$
$$\leq \hat{L}_{SL} = \prod_{i=1}^{K-1} \max_{\sigma_i \in [0,1]^{n_i}} \left\| \widetilde{\Sigma}_{i+1} V_{i+1}^\top \operatorname{diag}(\sigma_{i+1}) U_i \widetilde{\Sigma}_i \right\|_2$$

where $M_i = U_i \Sigma_i V_i^\top$, $\widetilde{\Sigma}_i = \Sigma_i$ if $i \in \{1, K\}$ and $\widetilde{\Sigma}_i = \Sigma_i^{1/2}$ otherwise.

**Theorem** Let $M_k$ be the matrix associated to the $k$-th linear layer, $u_k$ (resp. $v_k$) its first left (resp. right) singular vector, and $r_k = s_{k,2}/s_{k,1}$ the ratio between its second and first singular values. Then we have

$$\hat{L}_{SL} \leq \hat{L}_{AL} \prod_{k=1}^{K-1} \sqrt{(1 - r_k - r_{k+1}) \max_{\sigma \in [0,1]^{n_k}} \langle \sigma \cdot v_{k+1}, u_k \rangle^2 + r_k + r_{k+1} + r_k r_{k+1}}.$$

If the ratios $r_k$ are negligible, then

$$\hat{L}_{SL} \leq \hat{L}_{AL} \prod_{k=1}^{K-1} \max_{\sigma \in [0,1]^{n_k}} |\langle \sigma \cdot v_{k+1}, u_k \rangle| \quad \text{and} \quad \hat{L}_{SL} \approx \frac{\hat{L}_{AL}}{\pi^{K-1}}.$$

# EXPERIMENTS

## 7. Experiments

**Theoretical setting**

Figure: SeqLip in the ideal scenario

Figure: Synthetic function to train MLPs

**MLP**

| | Upper bounds | | | | Lower bounds | | |
|---|---|---|---|---|---|---|---|
| # layers | Frob. | AutoLip | SeqLip | Greedy SL | Data | Annealing | Grid Search |
| 4 | 648.2 | 33.04 | 21.47 | 21.47 | 4.36 | 4.55 | 6.56 |
| 5 | 4283.1 | 134.4 | 72.87 | 72.87 | 6.77 | 5.8 | 7.1 |
| 7 | 22341 | 294.6 | 130.2 | 130.2 | 5.4 | 5.27 | 6.51 |
| 10 | 7343800 | 19248.2 | 2463.44 | 2463.36 | 10.04 | 5.77 | 17.1 |

Figure: AutoLip and SeqLip for MLPs of various size

**CNN**

| | Upper bounds | | | | Lower bounds | |
|---|---|---|---|---|---|---|
| # layers | AutoLip | Greedy SeqLip | Ratio | Dataset | Annealing |
| 4 | 174 | 86 | 2 | 12.64 | 25.5 |
| 5 | 790.1 | 335 | 2.4 | 16.79 | 22.2 |
| 7 | 12141 | 3629 | 3.3 | 31.22 | 43.6 |
| 10 | $4.5 \cdot 10^6$ | $8.2 \cdot 10^5$ | 5.4 | 38.26 | 107.8 |

Figure: AutoLip and SeqLip for MNIST-CNNs of various size

**AlexNet**

| AutoLip | Greedy SeqLip |
|---|---|
| $3.62 \times 10^7$ | $5.45 \times 10^6$ |

Figure: AutoLip and SeqLip for AlexNet

## 8. Conclusion and perspectives

SeqLip can improve by a **large factor** the upper-bound given by AutoLip (up to a **factor 8** in real scenarios). However, these bounds remain **very large for vision networks**, and it is yet an open question to know how close these bounds are to the **real Lipschitz constant**.