[Reviews · NeurIPS 2020]

Review 1

Summary and Contributions: ***Update after author feedback*** I keep my opinion that this is a good paper, and recommend acceptance. I see this method as the "exact" oracle (exponential complexity) and it should allow to compare how tight are other methods without exponential complexity, in experiments with small networks. Precisely in the way the authors use it in Table 1. For this reason it is crucial that authors release the code on publication. Rebuttal partly addressed one of my questions regarding the general position property. Authors didn't address the following concern, that I rewrite here, but it should be easily fixed with minor rewriting. There is a slight mistake about previous work. The reference [10] also supports local Lipschitz estimation. The authors may want to fix this claim. *** This paper introduces an algorithm based on Mixed Integer Programming (MIP), that correctly computes the Lipschitz constant of a ReLU network in "General Position". The "General Position" assumption seems not restrictive as it holds with probability one with respect to the Lebesgue Measure. The correctness of the algorithms is obtained via a characterization of the Lipschitz constant as a supremum of the norm of the clarke generalized subdifferential, over points of differentiability. Some slightly different versions of such result have been presented before.

Strengths: The approach is theoretically well supported. Comparison with previous alternative techniques is extensive. In particular the method presented here allows a clear comparison as it provides the true parameters other methods try to estimate. It is relevant to the community.

Weaknesses: I think that the main limitation is the computational cost of the proposed algorithm, as mixed integer programming is a hard problem. As such, the proposed method won't probably scale to large networks used in industrial applications. However this drawback is clearly acknowledged and is is still useful for small networks or to compare and tightly asses the quality of other methods in a small benchmark. Also there is a slight mistake about previous work. The reference [10] also supports local Lipschitz estimation. The authors may want to fix this claim.

Correctness: Derivations seem correct. In particular one of the main issues, often overlooked in other papers, is the handling of the non-differentiability of ReLU networks and the lack of a chain-rule for such networks. In contrast, the paper handles them using the standard notion of the clarke subdifferential, making claims precise. The empirical methodology seems ok.

Clarity: The paper is really well-written and it was easy for me to follow throughout the main body. It presents the calims and assumptions clearly and with a nice style.

Relation to Prior Work: the paper places itself with respect to prior work and has appropriate citations. The main differences are highlighted.

Reproducibility: Yes

Additional Feedback: The only thing I feel is missing, is a discussion on how to verify if a network is in general position, and the computational complexity of such a method. In principle, one has to perturb the weights of the network with some noise e.g. gaussian, for it to be in general position. THis might affect the accuracy and other properties of the network, and it is not clear if adding noise would be a good idea for deeper networks.


Review 2

Summary and Contributions: This work computes the exact local Lipschitz constant of ReLU networks. Unlike previous work, such as FastLip or CLEVER, which relates the Lipschitz constant to the maximal dual norm of a network’s gradient, this approach instead relates that to the supremal norm of its generalised Jacobian matrix, so that the Lipschitz constant of vector-valued (e.g., in unsupervised learning) and nonsmooth (e.g., ReLU) networks, not just scalar-valued smooth ones, can be calculated.

Strengths: The authors claim that this is the first work to compute the exact local Lipschitz constant of ReLU networks, which could be used as a ground-truth to evaluate other Lipschitz estimators, and also make claims regarding the properties of networks such as the effect of regularisation during training (Figure 1, Section 7) in the experimental results.

Weaknesses: The empirical experiments (Table 1) show that LipMIP could be significantly slower than its alternatives - for example, it is 10^4 times slower than FastLip. This is understandable and somewhat expected because it computes the exact value, as in Section 4 the authors justify LipMIP’s exponential-time performance in worst-case. However, the advantage of its convex relaxation LipLP, which produces upper bounds and not exact values, over FastLip, which also provides upper bounds, is not obvious - both yield more or less similar relative error but the latter is much faster. Also, we can see that the size of the networks in this experiment is rather small, which means in reality for most cases LipMIP might only be able to produce upper bounds as it will need to be stopped early. Besides, the novelty of the work is not substantial. It is not new to encode networks using MIP. For instance, in robustness certification of neural networks there has been work involving MIP-encoding of networks, such as the Sherlock approach. Also, the subroutine for bound propagation here is the same as the formulation of FastLip. That said, though this work claims to compute the first exact value of local Lipschitz, the majority of the methodology behind it does follow previous threads.

Correctness: Technically speaking, the authors provide a sufficient condition over ReLU, referred to as general position ReLU networks, so that the generalised chain rule yields the generalised Jacobian. To this point, the feasible set of Equation (4) changes to that of Equation (7). Subsequently, they encode the general position ReLU networks using mixed-integer programming (MIP), i.e., LipMIP, via writing the network as a composition of affine, conditional, and switch operators. When the global lower and upper bounds for each input are known, these three operators are MIP-encodable thus so is the network itself. When encoding is done, extant MIP solvers can be utilised.

Clarity: The writing is good

Relation to Prior Work: Yes - this adds to a stream of research aiming at computing Lipschitz constants for neural network, where the authors claim they are the first to compute the exact local constant

Reproducibility: Yes

Additional Feedback: Suggestions to improve the paper: For the experimental results in Table 1 Section 7, is it possible to show the actual values of the calculated Lipchitz constants rather than just the relative errors of different estimators? Also, the authors mention that these results are for networks with various sizes, does that mean only the average values for different sizes are presented? Are there anything interesting regarding the comparison of these estimators on different sizes, as it can also be a hint of scalability behind? As for Figure 1, regarding the Lipschitz values during network training, is it statistically significant that the performance of LipSDP is better than LipLP, and the latter is better than FastLip? Besides, apart from the effect of regularisation during training, are there any other properties of networks that could be included to demonstrate the merit of an exact Lipschitz value so as to strengthen the contribution of the paper? Typos / errors: + Line 292, the citations of CLEVER and FastLip are exchanged, which could be misleading to readers. + Line 204, is the formula missing "( )”? + Line 236, … into the feasible set a la Equation 9 …?


Review 3

Summary and Contributions: This paper proposes an algorithm LipMIP for the exact computation of Lipschitz constants for ReLU networks. This algorithm works for ReLU networks that are in general position, a condition defined in the paper which only excludes a zero measure of ReLU networks. The algorithm invokes mixed-integer programming solvers and thus runs in exponential time in the worst case. But a provably efficient algorithm may not exist in theory: the paper also proves the ETH-hardness of estimating Lipschitz constants for ReLU networks, even for approximating within any factor that scales sublinearly with the input dimension.

Strengths: I like this paper a lot. Computing Lipschitz constants of a neural network is a fundamental problem and has potential application in GAN, adversarial robustness, generalization bounds. The algorithm is grounded by solid theory. The authors conduct experiments to compare the efficiency and accuracy of LipMIP against other estimators. The notion of "general position" is particularly interesting. Assuming this, backpropagation always returns an element of generalized Jacobian (in Clarke's sense) despite there is no chain rule for non-smooth functions in general. This interesting property is used for designing LipMIP and may be of independent interest.

Weaknesses: The algorithm can only work for Lipschitz constants with respect to L1 norm, Linf norm, or linear norms (Definition E.1). It seems to require fundamentally new ideas to generalize it to other norms such as L2 norm, because MIP cannot encode non-linear norms.

Correctness: The theorems look correct although I have not verified all the details.

Clarity: This paper is well-written, but there are minor flaws. In Line 41, "under certain norms" is very unclear. I didn't realize that the algorithm cannot work for L2 norm until I read Section 5. In Line 55, the hardness actually holds for factors that scales almost linearly with input dimension.

Relation to Prior Work: The related works section is well-written.

Reproducibility: Yes

Additional Feedback: Currently there is no running time gaurantees for LipMIP. It would be nice if the authors can provide explicit bounds for the number of variables and inequalities in the MIP instance. All the experiments are conducted on small networks. I hope the authors could discuss the limitation of this algorithm when scaling up to larger networks. ---------------------- I'm satisfied with the authors' response on the scalability issues, so I would like to maintain my score.


Review 4

Summary and Contributions: This paper provides a novel method to exactly compute the local Lipchitz constant of ReLU network via the mixed-integer programming strategies.

Strengths: Estimating the Lipchitz constant has received much attention in the machine learning community, which plays a vital role in adversarial robustness, WGAN, to name a few. This paper provides an interesting method to compute it starting from the supreme dual norm. Furthermore, they claim the chain rule holds for ReLU network under the general position assumption (ps: it is a very interesting finding, but I have some concerns which will elaborate later). Frankly, due to limited time, I just go through the proof and am not sure whether it is correct for section 3 in supplementary. Lastly, this work conducts extensive experiments to corroborate their theoretical finding based on Mix-integer programs.

Weaknesses: 1. Except for section 3, other theoretical findings/results seem rather standard. For example, the reformulation techniques involved in section 5 have been widely used in mixed-integer programs and even for certifying the adversarial robustness, e.g., SMT solver, nothing new. Also, theorem 1 can be easily obtained via several simple inductions. 2. For theorem 2, I am curious about whether the function w.r.t. ReLU network is subdifferential regular under the general position assumption? See Theorem 49 in [1] for details. Even for the seemingly simple function f(x) = (max(x,0)-1)^2 (e.g., square loss), it is not regular, see deep neural network example in section IV [2]. Thus, I am suspicious of why the general position assumption can fill the gap? Maybe you can use the one layer ReLU network as an example to showcase the gap? 3. From the computational viewpoint, MIP relies on the general-purpose solvers, which cannot scale well with the deep learning tasks. However, this point is minor to me. If you can answer my questions and concerns or clarify something (i.e., might have some misunderstanding about your works), I feel free to increase my score. [1] Rockafellar R T, Wets R J B. Variational analysis[M]. Springer Science & Business Media, 2009. [2] Li J, So A M C, Ma W K. Understanding Notions of Stationarity in Non-Smooth Optimization[J]. arXiv preprint arXiv:2006.14901, 2020.

Correctness: Not sure.

Clarity: Yes.

Relation to Prior Work: Yes.

Reproducibility: Yes

Additional Feedback: See above. ============================== Thanks for your response. Now, I am fine with your general position assumption and adjust my score.

[Author Response · NeurIPS 2020]

We thank the authors for their careful reading and helpful comments. We have addressed typos, minor mistakes with related work and other comments provided in the "Clarity" section of the reviews. We first summarize the main critiques raised by the reviewers. Reviewer #1 had concerns about the computational cost of our proposed algorithm and the verifiability of our "general position assumption". Reviewer #2 also raised concerns about the scalability of our approach and noted that much of our algorithmic methodology follows previous threads in neural net verification. Reviewer #3 would like us to stress that our proposed approach only holds for a narrow class of norms that excludes the L2 norm, and requested a discussion on scalability. Reviewer #4 commented that our results appear standard and asked a technical question regarding our general position assumption. They also noted the scalability concerns. We will proceed by addressing the shared concerns of our reviewers, and then respond to the individual suggestions/questions.

**On scaling to large networks:** As we acknowledge in the paper, the primary drawback to our technique is that MIP approaches have worst-case runtime that is exponential in the number of integer variables. In particular, LipMIP will require one integer variable for each ReLU unit that is "unstable" within a region in addition to at most one integer variable per input dimension; this depends on the network, but also the size of the region we wish to verify. As we scale to larger networks or very high-dimensional datasets, this quickly becomes intractable, though there is hope that the regularization techniques of [1] may provide a modest boost in performance by regularizing towards the stability of neurons. On very large networks, the precision of MIP-solvers may also become an issue. We are working on applying our framework in sub-networks to yield efficient bounds for very large models.

**On novelty of results:** Reviewers #2 and #4 correctly note that MIP-encoding of neural networks for robustness verification is well-studied. However the extension to Lipschitz-computation is novel and technical: it requires novel machinery to encode the backpropagation procedure as well as the careful theoretical analysis of sections 2 and 3 to prove that the optimization problem is well-founded and returns the correct answer. To the best of our knowledge, there is no extant result relating the Lipschitz constant to the generalized Jacobian, though if such a result exists, we will happily discuss and cite it. Further, our inapproximability result significantly extends what was known about the hardness of estimating Lipschitz constants of ReLU networks.

**R#1: "The only thing I feel is missing, is a discussion on how to verify if a network is in general position, and the computational complexity of such a method."** Good point. We believe that verifying whether or not a given network is in GP is actually computationally hard (See also related proofs on hardness of checking the Restricted Isometry property). There is an easy solution: an arbitrarily small random perturbation of the weights will create a GP network almost surely. This cannot significantly affect accuracy since floating point arithmetic already introduces a small amount of noise during inference, and the classification loss is typically Lipschitz with respect to network parameters. Further, randomly initialized networks that are trained using SGD will be in general position almost surely if infinitesimal noise is added to each gradient step. We will discuss this in the paper.

**R#2: Experimental questions/suggestions and "...are there any other properties of networks that could be included to demonstrate the merit of an exact Lipschitz value so as to strengthen the contribution of the paper?"** We will present a more complete version of Table 1 in future iterations. To clarify the caption for table 1: an architecture was fixed for each dataset and many initializations/trainings were evaluated to yield the average and variances. All results were statistically significant. LipLP will always do better than FastLip, at the cost of extra computation time. The relative performance of LipSDP largely depends on the input dimension, with smaller input dimension yielding much tighter results. We also note that our technique may be easily extended to other piecewise-linear activation functions, such as LeakyReLU's or Hard Tanh's. These may be used to verify the conjectures made in [2] regarding the effect activation functions have on Lipschitz constants. We will include these results in future iterations.

**R #4: "For theorem 2, I am curious about whether the function w.r.t. ReLU network is subdifferential regular under the general position assumption?"** Great question. Subdifferential regularity (SR), roughly speaking, allows the Clarke subdifferential to adopt the nice properties of the subdifferential for convex functions, such as the chain rule. SR is different from our General Position (GP) definition and not very useful for ReLU nets. For example, the authors of [3] (sec 5.1) show that even $-|x|$ is not SR at 0. We can consider the 1-layer neural network $f(x) = \sigma(x+1) - \sigma(2x)$, which locally looks like $-|x|$ at 0 and thus is not subdifferentially regular at 0. Observe that $f$ here is a ReLU network in GP. Hence, our GP results seem to be similar to a notion of stratification, where the strata are the ReLU kernels. We would like to argue that this generality of GP (versus SR) is another example of technical novelty of our work.

[1] Kai Y Xiao, Vincent Tjeng, Nur Muhammad Shafiullah, and Aleksander Madry. Training for faster adversarial robustness verification via inducing relu stability. *arXiv preprint arXiv:1809.03008*, 2018.

[2] Mina Basirat and PETER ROTH. L* relu: Piece-wise linear activation functions for deep fine-grained visual categorization. In *The IEEE Winter Conference on Applications of Computer Vision*, pages 1218–1227, 2020.

[3] Damek Davis, Dmitriy Drusvyatskiy, Sham Kakade, and Jason D Lee. Stochastic subgradient method converges on tame functions. *(FOCS) Foundations of computational mathematics*, 20(1):119–154, 2020.


[Meta-Review · NeurIPS 2020]

This is an interesting and thorough paper on computing the Lipschitz constant of deep networks. On the positive side, the paper is very thorough, both in terms of theory and practice; personally, I was surprised to find it even included a hardness result, and feel that this paper has many things for many researchers. I look forward to seeing this paper appear, and support the authors in further investigations. --- Minor comment. Some feedback which may be interesting for future directions. Some of the negative points in discussion were the hopelessness of exact computation, and the appearance of MIP in prior work in this field; perhaps there are some interesting relaxed approaches, based on the insights in this paper? Overall, reviews enjoyed the paper, I'm just trying to be helpful for future work.